# Unsupervised Emergence of Spatial Structure from Sensorimotor Prediction

## Abstract

Despite its omnipresence in robotics application, the nature of spatial knowledge and the mechanisms that underlie its emergence in autonomous agents are still poorly understood. Recent theoretical work suggests that the concept of space can be grounded by capturing invariants induced by the structure of space in an agent's raw sensorimotor experience. Moreover, it is hypothesized that capturing these invariants is beneficial for a naive agent trying to predict its sensorimotor experience. Under certain exploratory conditions, spatial representations should thus emerge as a byproduct of learning to predict. We propose a simple sensorimotor predictive scheme, apply it to different agents and types of exploration, and evaluate the pertinence of this hypothesis. We show that a naive agent can capture the topology and metric regularity of its spatial configuration without any a priori knowledge, nor extraneous supervision.

## 1 Introduction

Space appears to be a pervasive concept in our perception of the world, and as such plays a central role in most artificial perception systems, in particular in computer vision and robotics applications. Yet its fundamental nature and the mechanisms that could lead to its emergence in an artificial system still remain poorly understood, despite some noteworthy contributions (Kant, 1781; Poincaré, 1895; Nicod, 1924). In most cases, the problem is circumvented by implementing prior knowledge in the system regarding the structure of space, and how motor and sensory information convey spatial properties (for instance through a kinematics model (Siciliano & Khatib, 2016), or a sensor model (Cadena et al., 2016)). In more recent years, and with the developments of machine learning techniques, approaches with less hand-engineered priors developed to solve spatial tasks ((Kahn et al., 2017; Quillen et al., 2018; Levine et al., 2018; Smolyanskiy et al., 2017) to name a few). However they tend to set aside the specificity of spatial experiences in favor of a global assessment of the agent's performance, leaving the question of the origin and structure of spatial knowledge largely open.

Nevertheless, recent work presented in (Laflaquière et al., 2018), directly addresses the grounding of spatial knowledge. It gives a theoretical perspective on how a naive agent could build a notion of spatial configuration, in an unsupervised way. The approach takes inspiration from philosophical ideas formulated more than a century ago by H.Poincaré (Poincaré, 1895), and from the more recent Sensori-Motor Contingencies Theory (SMCT) which puts forward an unsupervised sensorimotor grounding of perception (O'Regan & Noë, 2001). It describes how space naturally induces specific invariants in any situated agent's sensorimotor experience, and how these invariants can be autonomously captured to improve the compactness of internal representations and the prediction of sensorimotor experiences.

In this paper, we propose a practical evaluation of this theoretical work. We introduce a simple self-supervised learning scheme in which an agent learns to predict its sensory experience, given its motor state. We analyze the encoding of the motor experience produced this way, and in particular how it is impacted by different types of exploration of the environment. More precisely, we show that by experiencing consistent sensorimotor transitions in a moving environment, a naive agent can learn to encode its motor state in such a way as to capture the topological structure and metric regularity of its external position, without the need for additional a priori knowledge.

## 2 RELATED WORK

The problem of building spatial representations of some sort is addressed in a multitude of machine learning papers. Due to space constraints, we cannot give a complete overview, but will only highlight the most relevant to this work.

The problem of spatial knowledge acquisition is often conceptualized in a supervised or reinforcement learning (RL) framework. The goal is to build an internal representation of an agent's position and displacement in an allocentric frame. Many approaches have been proposed to represent localization in the environment as place cells (Arleo et al., 2001; Weiller et al., 2010; Stachenfeld et al., 2017). These representations are typically grounded in the sensory space, making them brittle to environmental changes. They also rely on specifically designed priors about how to form and compare prototypical sensory states. An extreme case would of course be SLAM-type approaches where the whole spatial structure of the problem is hard-coded a priori in the robot (Cadena et al., 2016). Taking inspiration from biology, other approaches rather focus on representing displacements via grid cells (Banino et al., 2018; Cueva & Wei, 2018). They typically rely on the prior definition of place cells, or on the direct access to the agent's position in the world during training.

Many works focus on building state representation for RL. Although not necessarily spatial in nature, these representations are often optimized to solve spatial tasks. They typically rely on priors such as physical properties of the world (Jonschkowski & Brock, 2015), controllability of the representation (Watter et al., 2015), or disentanglement of the representation (Thomas et al., 2017). A large class of RL-based approach also aims at solving intrinsically spatial tasks in an end-to-end fashion. This is for instance the case of the influential Atari playing agent (Mnih et al., 2013), or of (Mirowski et al., 2016) where auxiliary tasks are used to enforce some spatial properties (loop closing). Recently, (Eslami et al., 2018) proposed a network able to generate perspective-conditioned images of 3D scenes, given explicit spatial perspective inputs. These work are however focused on solving a task, and do not investigate how spatial knowledge might be internally encoded and used by the agent.

A last class of work rely more on unsupervised or self-supervised mechanisms to build spatial representations or solve spatial tasks. This is for instance the case in (Ha & Schmidhuber, 2018; Wayne et al., 2018), where compression and self-prediction are used to build a world model helping an agent to navigate in its environment. Intrinsic curiosity has also been used to learn to control an agent moving in an environment, even in the absence of reward (Pathak et al., 2017). More related to this work, sensorimotor prediction has also been considered as a mechanism to integrate motor sequences into displacement-like representations (Ortiz & Laflaquière, 2018).

The work presented in this paper differs from these related works in that we study the emergence of the concept of egocentric position. A naive agent with only access to its sensorimotor flow indeed has to discover that the motor configurations it produces correspond to external spatial configurations, with their own topology and (Euclidean) metric. We cast the problem outside of the typical reinforcement learning framework, by not considering any extraneous reward, and by not assuming the existence of any spatial prior. We instead consider learning of a sensorimotor predictive model as the only driving force in building spatial representations. Moreover, we explicitly study how representations with spatial properties can be derived from the motor space. As such, this paper is more in line with theoretical work addressing the problem of space perception from the SMCT perspective (Laflaquiere et al., 2012; Laflaquière et al., 2015; Terekhov & ORegan, 2016).

## 3 PROBLEM SETUP

In this work, we study how a naive agent can build an internal representation of its external position in an egocentric frame. Inspired by Poincaré's insight (Poincaré, 1895), and borrowing from differential geometry (Philipona et al., 2003), we look at the agent's experience as a sensorimotor mapping parametrized by the state of the environment. We assume that an agent's instantaneous sensory and motor states are respectively defined by vectors $\mathbf{s} \in \mathbb{R}^{N_s}$ and $\mathbf{m} \in \mathbb{R}^{N_m}$. The motor states can be seen as static joint configurations, or as proprioceptive readings in the case the agent is not controlled in position. No additional assumption is made regarding the way sensorimotor information is encoded. The unknown sensorimotor mapping induced by the environment is assumed continuous and denoted $\phi_\epsilon$, such that $\mathbf{s} = \phi_\epsilon(\mathbf{m})$, where $\epsilon \in \mathbb{R}^{N_\epsilon}$ is the state of the environment. The mapping $\phi_\epsilon$ can be seen as describing how "the world" transforms changes in motor states into

changes in sensory states, and thus encapsulates the structure of the external space in which the agent is embedded. Finally we denote $\mathbf{p} \in \mathbb{R}^{N_p}$ the unknown spatial position of the agent's sensor in an egocentric frame of reference. The problem we address in this work is the one of building an internal representation $\mathbf{h}$ which captures the properties of $\mathbf{p}$. Indeed, $\mathbf{p}$ has some topological and metric properties that the naive agent cannot directly access. They need to be derived from sensorimotor experiences. It has been shown that space induces, through $\phi_\epsilon$, specific sensorimotor invariants that an agent can capture to discover the structure of $\mathbf{p}$ (Laflaquière et al., 2018; Laflaquiere et al., 2013).

A first kind of invariants concerns the *topology* of the sensor position $\mathbf{p}$. Assuming a rich enough environment and sensory apparatus, there exists a homeomorphic relation between the sensor position and the sensory manifold generated by exploring the motor space. Intuitively, for all environment states $\epsilon$, motor changes which generate small sensor displacements are associated with small sensory changes:

$$\forall \epsilon, \text{ if } (\mathbf{m}_i, \mathbf{m}_{i'}) \text{ is such that } |\overrightarrow{\mathbf{p}_i \mathbf{p}_{i'}}| \ll \mu \text{ then } |\overrightarrow{\phi_\epsilon(\mathbf{m}_i)\phi_\epsilon(\mathbf{m}_{i'})}| = |\overrightarrow{\mathbf{s}_i \mathbf{s}_{i'}}| \ll \mu, \qquad (1)$$

where $|.|$ denotes the Euclidean norm, and $\mu$ is a small value. This means that the topology of the sensor's position is accessible to the agent via the sensorimotor flow. Note that in the case of a redundant motor system, multiple motor states $\mathbf{m}$ lead to the same position $\mathbf{p}$, and thus to identical sensory states ($|\overrightarrow{\mathbf{s}_i \mathbf{s}_j}| = 0$), for any state of the environment $\epsilon$. Based on this sensory equivalence, the agent can discover that the manifold of positions $\mathbf{p}$ has a lower dimension than its motor space. This space-induced topological invariant should be accessible to the agent under **condition I**: the agent should experience *consistent* sensorimotor transitions such that the state of the environment $\epsilon$ does not change during the transition (see Eq. (1)).

A second kind of invariant concerns the *metric regularity* of the sensor position $\mathbf{p}$. Assuming the environment can move relative to the agent, multiple motor changes associated with the same external displacement of the sensor can generate the same sensory changes, depending on the position of the environment. Consider for instance that the agent looks at a bottle that can be at different locations. Then the same sensory change, corresponding to moving the sensor (camera) from the bottle's top to its bottom, can be generated by different motor changes depending on where the bottle is located relatively to the agent. All these motor changes actually correspond to the same external displacement (change of position) of the sensor. This sensory equivalence of motor changes associated with the same displacement holds for all environmental states (the bottle in the example can be replaced with any other object):

$$\forall \epsilon, \text{ if } (\mathbf{m}_i, \mathbf{m}_{i'}, \mathbf{m}_j, \mathbf{m}_{j'}) \text{ is such that } |\overrightarrow{\mathbf{p}_i \mathbf{p}_{i'}} - \overrightarrow{\mathbf{p}_j \mathbf{p}_{j'}}| \ll \mu$$
$$\text{then } \exists \delta(\epsilon) : |\overrightarrow{\phi_\epsilon(\mathbf{m}_i)\phi_\epsilon(\mathbf{m}_{i'})} - \overrightarrow{\phi_{\epsilon+\delta(\epsilon)}(\mathbf{m}_j)\phi_{\epsilon+\delta(\epsilon)}(\mathbf{m}_{j'})}| = |\overrightarrow{\mathbf{s}_i \mathbf{s}_{i'}} - \overrightarrow{\mathbf{s}_j \mathbf{s}_{j'}}| \ll \mu, \qquad (2)$$

where $\delta(\epsilon)$ represents a displacement of the environment. This means that the spatial equivalence of motor changes, and thus the metric regularity of the sensor position $\mathbf{p}$ are accessible to the agent via the sensorimotor flow. This space-induced metric invariant should be accessible to the agent under **condition II**: the agent should experience *displacements of the environment* in-between sensorimotor transitions (see Eq. (2)). This ensures the agent has the opportunity to experience the same sensory changes with different motor changes.

It has been shown (Laflaquière et al., 2018) that it is theoretically possible to build a motor representation $\mathbf{h}$ which respects space-induced invariants by: i) having similar representations $\mathbf{h}$ for motor states $\mathbf{m}$ associated with similar sensory states $\mathbf{s}$, and ii) having similar representation changes $\overrightarrow{\mathbf{hh'}}$ for motor changes $\overrightarrow{\mathbf{mm'}}$ associated with similar sensory changes $\overrightarrow{\mathbf{ss'}}$. We hypothesize that building such a representation is beneficial to a naive agent learning to predict its own sensorimotor experience. Indeed, these invariants hold for all states of the environment. They should thus act as a prior over sensorimotor transitions not yet experienced, improving this way the predictive capacity of the system. As a consequence, sensorimotor prediction should naturally lead to the emergence of a representation $\mathbf{h}$ capturing the topology and metric regularity of the egocentric spatial configuration $\mathbf{p}$. Given the topological and metric relations we described, any $\mathbf{h}$ which is an affine transformation of the ground-truth position $\mathbf{p}$ would be an optimal solution, as such a transformation preserves topology and distance ratios.

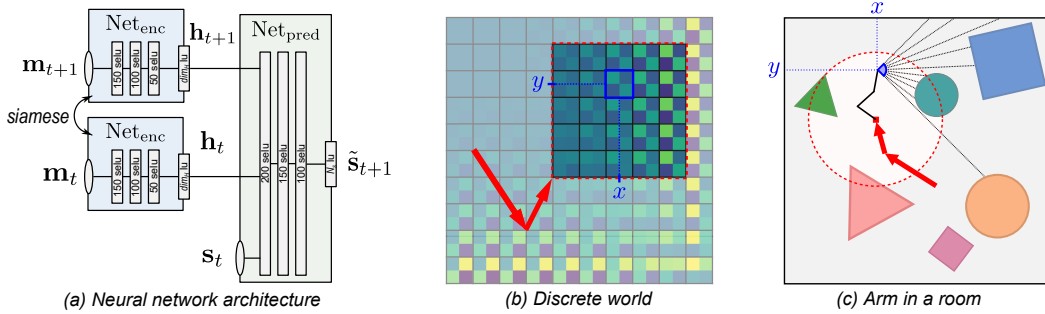

*(a) Neural network architecture*  *(b) Discrete world*  *(c) Arm in a room*

Figure 1: (a) The neural network architecture featuring two siamese instances of the $\text{Net}_{enc}$ module, and the $\text{Net}_{pred}$ module. (b-c) Illustrations of the discrete world and arm in a room simulations. The part of the environment accessible to the agent for the current translation of the environment is framed in dotted red. The agent's current position $[x, y]$ in this frame is displayed in blue. Red arrows illustrate the type of translation the environment can undergo. They are equivalent to a displacement of the agent's base in a static environment. (Best seen in color)

## 4 EXPERIMENTS

### 4.1 SENSORIMOTOR PREDICTIVE NETWORK ARCHITECTURE

We propose a simple sensorimotor predictive architecture to test the validity of the hypotheses laid out in Sec. 3. The architecture is made of two types of modules: i) $\text{Net}_{enc}$, a Multi-Layer Perceptron (MLP) taking a motor state $\mathbf{m}_t$ as input, and outputting a motor representation $\mathbf{h}_t$ of dimension $dim_H$, and ii) $\text{Net}_{pred}$, a MLP taking as input the concatenation of a current representation $\mathbf{h}_t$, a future representation $\mathbf{h}_{t+1}$, and a current sensory state $\mathbf{s}_t$, and outputting a prediction for the future sensory state $\tilde{\mathbf{s}}_{t+1}$. As illustrated in Fig. 1, the overall network architecture connects a predictive module $\text{Net}_{pred}$ to two siamese copies of a $\text{Net}_{enc}$ module, ensuring that both motor states $\mathbf{m}_t$ and $\mathbf{m}_{t+1}$ are consistently encoded using the same mapping. The self-supervised sensorimotor predictive task of the network consists in minimizing the Mean Squared Error between the prediction $\tilde{\mathbf{s}}_{t+1}$ and the ground truth $\mathbf{s}_{t+1}$. No explicit component is added to the loss regarding the representation $\mathbf{h}$. Unless stated otherwise, the dimension $dim_H$ is arbitrarily set to 3 for the sake of visualization. A thorough description of the network is proposed in Appendix A, alongside a description of the training procedure.

### 4.2 TESTING HYPOTHESIS

We introduce a measure of how well $\mathbf{h}$ approximates the ground truth position $\mathbf{p}$ of the sensor. Unfortunately, the two cannot be compared directly, as there might exists an affine transformation between the two (see Sec.3). For a given set of representations $H = \{\mathbf{h}_i\}$ and their ground-truth counterparts $P = \{\mathbf{p}_i\}$, we thus perform a linear regression between $P$ and $H$ to estimate and compensate for this potential affine transformation (and similarly between $H$ and $P$). After this compensation, pairwise distances in $P$ and $H$ can be compared to evaluate how much the structure of the two sets differ. We thereby define two measures of dissimilarity as:

$$Q_p = \frac{1}{N^2}\sum_i^N\sum_j^N \frac{|D_{ij}^{H\rfloor P} - D_{ij}^P|}{\max_{kl}(D_{kl}^P)} \ , \ Q_h = \frac{1}{N^2}\sum_i^N\sum_j^N \frac{|D_{ij}^{P\rfloor H} - D_{ij}^H|}{\max_{kl}(D_{kl}^H)}, \tag{3}$$

where $N$ is the number of samples, $D_{ij}^X$ denotes the distance between samples $i$ and $j$ in the set $X$, $H\rfloor_P$ denotes the set $H$ after projection in the space of $P$ by affine compensation, and $P\rfloor_H$ the set $P$ after a similar projection in the space of $H$. The errors are normalized by the maximal distance in the projection space in order to avoid undesired scaling effects. In order to rigorously compare dissimilarity measures between experiments, sets $H$ and $P$ are always generated by sampling the motor space in a fixed and regular fashion (see Fig. 3).

In order to test the hypotheses of Sec. 3, three types of exploration of the environment are considered:

- **Inconsistent transitions in a moving environment**: A baseline scenario in which the spatiotemporality of sensorimotor experiences is not respected. Pairs of states $(\mathbf{m}_t, \mathbf{s}_t)$ and $(\mathbf{m}_{t+1}, \mathbf{s}_{t+1})$ are generated by randomly sampling the motor space, while the environment's position randomly changes between $t$ and $t + 1$, thereby breaking conditions I and II. It is akin to what a typical passive and non-situated (not in a continuous interaction with the world) agent would experience. We will refer to this type of exploration as **MTM** (Motor-Translation-Motor).
- **Consistent transitions in a static environment**: A situated scenario in which the agent can experience the spatiotemporal consistency of its sensorimotor transitions. Pairs of states $(\mathbf{m}_t, \mathbf{s}_t)$ and $(\mathbf{m}_{t+1}, \mathbf{s}_{t+1})$ are generated by randomly sampling the motor space, while the environmental stays static, thereby fulfilling condition I, but not condition II. We will refer to this type of exploration as **MM** (Motor-Motor).
- **Consistent transitions in a moving environment**: A situated scenario in which the agent can experience consistent transitions and displacements of the environment. Pairs of states $(\mathbf{m}_t, \mathbf{s}_t)$ and $(\mathbf{m}_{t+1}, \mathbf{s}_{t+1})$ are generated by randomly sampling the motor space, while the environment's position can randomly change after the two pairs have been collected, thereby fulfilling conditions I and II. We will refer to this type of exploration as **MMT** (Motor-Motor-Translation).

Additional details about the sampling process are available in Appendix A. According to hypotheses of Sec. 3, the motor representation built via a MTM exploration should capture no spatial invariants, the one built via a MM exploration should capture spatial topology, and the one built via MMT epxloration should capture both spatial topology and metric regularity.

### 4.3 AGENT-ENVIRONMENT SETUPS

We propose two different simulated agent-environment setups to test our hypotheses:

*Discrete world*: An artificial setup designed to provide a simple and fully controlled sensorimotor interaction. As illustrated in Fig. 1, the environment consists in a grid world of size $10 \times 10$, in a sensory state $\mathbf{s}$ of dimension $N_s = 4$ is associated which each cell. Each of these 4 sensory dimensions is set to be a random order 3 polynomial of the position in the grid. This ensures a continuous sensory experience when moving in the environment. The agent can freely explore a section of the environment of size $5 \times 5$. It can generate a 3D motor state $\mathbf{m} = [m_1, m_2, m_3]$, to which a sensor position $\mathbf{p} = [x, y] = [\sqrt{m_1}, \sqrt{m_2}]$ is associated. This arbitrary mapping is purposefully non-linear, and presents a superfluous motor command $m_3$. These properties will be useful to study how the final structure of $\mathbf{h}$ might differ from $\mathbf{m}$. Because of this non-linearity and the discrete nature of the positions $\mathbf{p}$, the agent can only sample its motor space in a non-linear fashion (see Fig. 3). When in a position $\mathbf{p}$, the agent receives the corresponding 4D sensory input $\mathbf{s}$. The environment can translate, effectively moving the workspace of the agent. Finally, the environment acts as a torus, which means that the sensor appears on the other side of the environment when exploring beyond its limits.

*Arm in a room*: A more complex scenario of an arm exploring a room, implemented using the Flatland simulator (Caselles-Dupré et al., 2018). The environment is a 2D square room of width 12 units with walls, randomly filled with simple geometric objects (squares, circles, triangles) with random properties (number, size, position, color) (see Fig. 1). The agent is a three-segment arm, each of length 1 unit, with one motor at each joint to control the relative orientation of the segment in $[-\pi, \pi]$ radians. The arm's tip is equipped with an array of 10 distance sensors. The orientation of the sensor in the world is fixed, such that the sensor only translates in the environment. The environment can translate relatively to the arm's base, and the agent's sensor cannot explore beyond the walls.

## 5 RESULTS

We evaluate the two experimental setups on the three types of exploration. Each simulation is run 10 times, with all random parameters drawn independently on each trial. During training, the dissimilarity measures $Q_p$ and $Q_h$ are evaluated on a fixed regular sampling of the motor space, so that they can be compared consistently between epochs. The evolution of the loss, $Q_p$, and $Q_h$ during training is displayed in Fig. 2. Additionally, Fig. 3 shows the final representation of the same regular motor sampling, for one randomly selected trial of each simulation. The corresponding ground truth positions $P$, as well as the affine compensations from the representation space to the

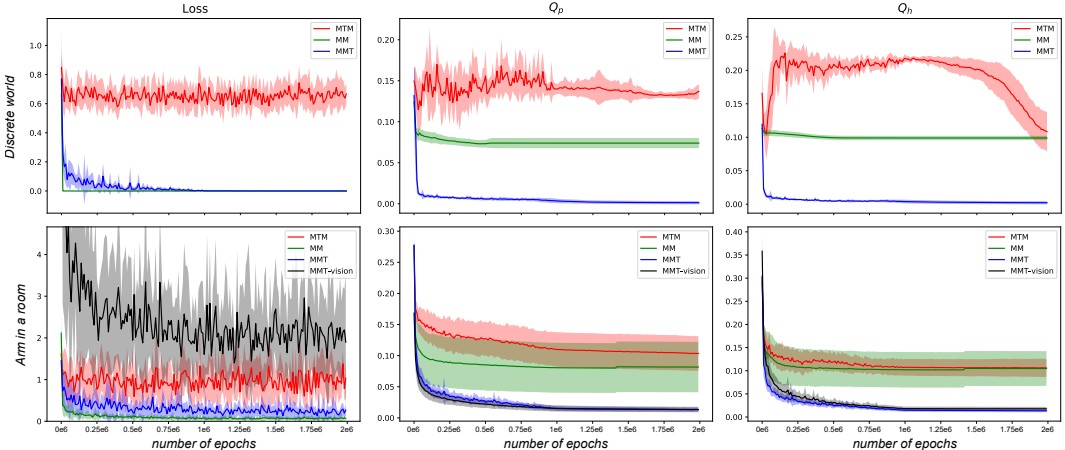

Figure 2: Evolution of the loss and the dissimilarity measures $Q_p$ and $Q_h$ during training for both setups, and for the three types of exploration. The displayed means and standard deviations are computed over 10 independent trials. (Best seen in color)

position space $H \rfloor_P$ (and conversely $H$ and $P \rfloor_H$ in the representational space) are also displayed. Below we present a qualitative analysis of the results and interpret them with regard to the initial hypotheses laid out in Sec.3. A more quantitative and detailed analysis of the results is provided in Appendix B.

## 5.1 MTM EXPLORATION

In the MTM exploration case, the loss, and measures $Q_p$ and $Q_h$ stay at relatively high values throughout the training, for both the discrete world and arm in a room simulations (see Fig. 2). Such a high loss indicates that the agent is unable to learn a good sensorimotor predictive mapping. This is expected as the environment moves during each transition. The current pair $(\mathbf{m}_t, \mathbf{s}_t)$ is therefore not informative to predict $\mathbf{s}_{t+1}$. Moreover, the high final $Q_p$ and $Q_h$ values seem to indicate that the structure of the representation $\mathbf{h}$ significantly differs from the ground truth position $\mathbf{p}$ of the agent's sensor, even after affine compensation. This is confirmed in Fig. 3 where we can see that the structure of $\mathbf{h}$ does not match the one of $\mathbf{p}$, metric-wise or even topology-wise. In particular, redundant motor states $\mathbf{m}$ corresponding to the same sensor position $\mathbf{p}$ are represented by different $\mathbf{h}$. When conditions I and II are not fulfilled, the motor representation $\mathbf{h}$ built by the predictive network thus converges to an arbitrary structure which does not capture the topology, and a fortiori the metric regularity, of the spatial configuration $\mathbf{p}$.

## 5.2 MM EXPLORATION

In the MM exploration case, both setups show a loss which quickly drops to very small values, while $Q_p$ and $Q_h$ rapidly decrease before stabilizing around high but significantly smaller values than in the MTM case (see Fig. 2). This sharp drop of the loss indicates that the network is able to learn an accurate sensorimotor predictive mapping. This is expected as the agent consistently explores a static environment; it can therefore easily map each motor state $\mathbf{m}_{t+1}$ to its corresponding sensory state $\mathbf{s}_{t+1}$. On the other hand, the still high but significantly lower values of $Q_p$ and $Q_h$ seem to indicate that the motor representation $\mathbf{h}$ built by the network is more similar to the ground truth position $\mathbf{p}$. In Fig. 3, the corresponding plots show that, after affine compensation, $\mathbf{h}$ indeed displays the same topology as $\mathbf{p}$ (best seen in the ground-truth position space). In particular, redundant motor states $\mathbf{m}$ corresponding to the same sensor position $\mathbf{p}$ are represented by the same $\mathbf{h}$. The representation thus effectively reduces the dimension of the motor manifold (3D) to match the dimension of the manifold of positions (2D). Yet, we can also see that the representation manifold is not necessarily flat, and does not match perfectly the flat affine projection of the positions in the representational space. This phenomenon is discussed in more details in Appendix B, but leads to $Q_h$ being less useful than $Q_p$ to assess the quality of the representation. This is particularly visible in Fig. 2

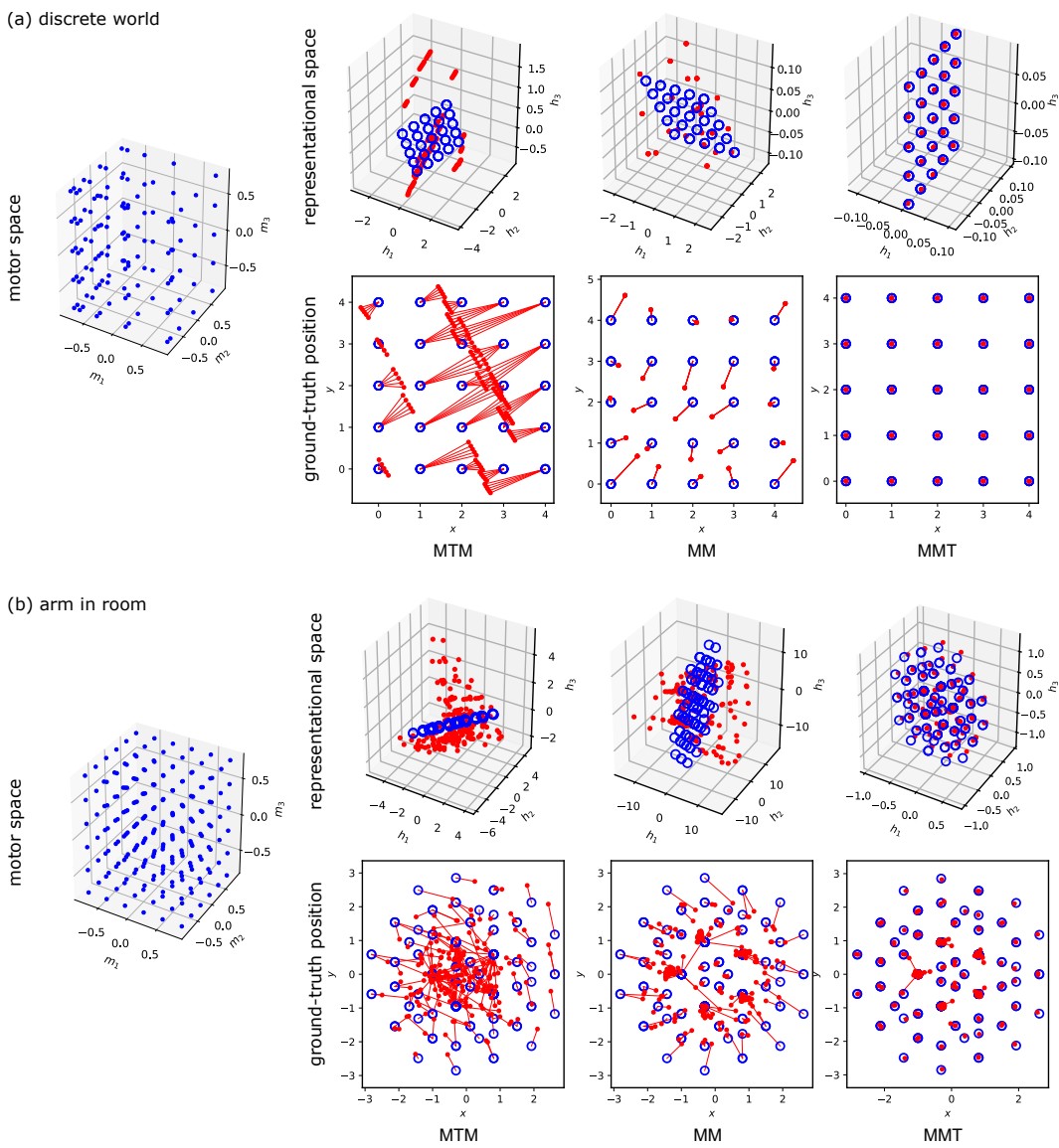

Figure 3: Visualization of the normalized regular motor sampling **m** (blue dots), its encoding **h** in the representational space (red dots), and the corresponding ground-truth position **p** (blue circles) for the three types of exploration, and for both the discrete world (a) and arm in a room (b) setups. Both the representations and the positions are also projected into each other's space via an optimized affine transformation. Lines have been added in the ground-truth position space to visualize the distances between the projections of **h** and their ground-truth counterparts **p**. (Best seen in color)

where $Q_h$ evolves towards similar final values in both the MTM and MM cases. It shows that $Q_h$ does not capture the structural difference of **h** induced by the two types of exploration, whereas $Q_p$ does. When condition I is fulfilled, the motor representation **h** built by the predictive network thus captures the topology of the external sensor position **p**. This is because the experience of consistent sensorimotor transitions in the environment induces (spatial) topological invariants which are captured by the predictive network (see Sec. 3).

### 5.3 MMT EXPLORATION

In the MMT exploration case, the loss, $Q_p$, and $Q_h$ all decrease and stabilize around very small values, for both setups (see Fig. 2). This loss drop indicates that the network is able to learn an

accurate sensorimotor predictive mapping. This is expected, as the agent experiences consistent sensorimotor transitions and the environment moves only between transitions. Knowing $\mathbf{m}_t$, $\mathbf{s}_t$, and $\mathbf{m}_{t+1}$, it is thus possible to predict the next sensory state $\mathbf{s}_{t+1}$. Moreover, the small final values of $Q_p$ and $Q_h$ indicate that the motor representation $\mathbf{h}$ built by the network is structurally very similar to the ground truth position $\mathbf{p}$. In Fig. 3, the corresponding plots indeed show that, after affine compensation, the motor representation $\mathbf{h}$ almost perfectly matches the position $\mathbf{p}$ (best seen in the ground-truth position space). They display the same topology and metric regularity, which shows that there exists a simple affine transformation between the two. Contrarily to the MM case, the 2D representation manifold even tends towards a flat surface. This phenomenon is further discussed in Appendix B. When conditions I and II are fulfilled, the motor representation $\mathbf{h}$ built by the predictive model thus captures both the topology and metric regularity of the external sensor position $\mathbf{p}$. This is because the experience of consistent sensorimotor transitions in the moving environment induces (spatial) topological and metric invariants which are captured by the predictive network (see Sec. 3).

We performed an additional experiment in order to evaluate the robustness of these result with respect to the complexity of the sensorimotor mapping and the dimension of the representational space. The arm is now equipped with a RGB camera of resolution 16 pixels, and $dim_H$ is set to 25 instead of 3. The results, displayed in black in Fig. 2, are qualitatively identical to the initial setup. This indicates that the emergence of the spatial-like structure in the representation $\mathbf{h}$ is insensitive to the complexity of the sensory input, and to the dimensionality of the representational space. These results are discussed at length in Appendix B.3.

## 6 CONCLUSION

We have proposed a practical evaluation of the unsupervised grounding of spatial structure via sensorimotor prediction. Using a simple neural network architecture with a motor state encoding module, we showed how sensorimotor prediction shapes motor representations such as to capture the topology and metric regularity of the external position of a sensor in an egocentric frame. In particular, we designed different types of exploration to induce spacial invariants, and showed that they were captured in the representation. Our results show that a naive agent needs to experience consistent sensorimotor transitions in order to discover the topology of space, and displacements of the environment in order to discover the metric regularity of space. Finally, they indicate that the pressure for accurate sensorimotor prediction is sufficient for these spatial properties to be captured. Intuitively, it appears that space-induced sensorimotor invariants drive motor states and motor transitions leading to similar sensory experiences to be represented similarly. Note however that the predictive module $\mathrm{Net}_{\mathrm{pred}}$ should, in theory, be able to learn an appropriate mapping regardless of the structure of the motor representation (assuming no loss of information). Understanding the precise mechanisms which drive the representation to capture sensorimotor invariants during training is thus an interesting question to investigate.

The results presented in this work suggest that no particular prior about space, nor about the properties of an agent's sensorimotor apparatus is required for the acquisition of spatial knowledge. By its very structure, space indeed induces invariants in an agent's sensorimotor experience, and those can be discovered by exploring the environment. This approach also suggests that action is essential in such an endeavor. It is of course necessary to explore the environment, but more fundamentally to discover spatial invariants. The results presented in this work would indeed be impossible to achieve with an agent processing only its sensory flow. Following Poincaré's intuition, the spatial-like representations $\mathbf{h}$ even originate from the motor space itself, whose internal structure is reshaped by sensorimotor experiences. Indirectly, this approach thus casts some doubts on purely passive and observational approaches of the problem of spatial knowledge acquisition.

Many problems need to be addressed before such an approach can be applied to more complex systems. So far, only translations of the sensor and the environment have been considered. The impact of adding rotations during exploration still needs to be studied theoretically, as the looped metric relations they induce cannot straightforwardly be represented in an Euclidean space. Additional theoretical work is also required to understanding how multiple spatial representations associated with multiple independent sensors can be merged to represent a global spatial configuration of the agent. Finally, the approach needs to be extended to characterize the environment's spatial configuration in the same egocentric frame, especially if it contains independent objects.

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

## A NEURAL NETWORK ARCHITECTURE AND TRAINING

*Neural network:* The sensorimotor predictive architecture is composed of two types of module: $\text{Net}_{\text{enc}}$ and $\text{Net}_{\text{pref}}$. The $\text{Net}_{\text{enc}}$ module projects a motor state $\mathbf{m}_t$ onto a representation $\mathbf{h}_t$ of dimension $dim_H$. It is a fully connected Multi-Layer Perceptron (MLP) with three hidden layers of size $(150, 100, 50)$, with SeLu activation functions (Klambauer et al., 2017), and a final output layer of size $dim_H$ with linear activation functions. This last layer corresponds to the representational space that is analyzed in this work. The $\text{Net}_{\text{pred}}$ module takes as input the concatenation $(\mathbf{h}_t, \mathbf{h}_{t+1}, \mathbf{s}_t)$ of a current motor representation, a future motor state representation, and a current sensory state, and outputs a prediction $\tilde{\mathbf{s}}_{t+1}$ of the future sensory state. It is a fully connected MLP with three hidden layers of size $(200, 150, 100)$, with SeLu activation functions, and a final output layer of size $N_s$ with linear activation functions. As illustrated in Fig. 1, the overall network architecture connects the predictive module $\text{Net}_{\text{pred}}$ to two siamese copies of the $\text{Net}_{\text{enc}}$ module, ensuring that both motor states $\mathbf{m}_t$ and $\mathbf{m}_{t+1}$ are consistently encoded using the same mapping.

*Loss minimization:* We define a simple self-supervised sensorimotor predictive objective for the network. The loss function to minimize is defined as the Mean Squared Error between the prediction $\tilde{\mathbf{s}}_{t+1}$ and the ground truth $\mathbf{s}_{t+1}$:

$$\text{Loss} = \frac{1}{K} \sum_{k=1}^{K} |\tilde{\mathbf{s}}_{t+1}^{(k)} - \mathbf{s}_{t+1}^{(k)}|^2,$$

where $k$ denotes a sample index, $K$ is the number of samples, and $|.|$ denotes the Euclidean norm. No particular component is added to the loss function regarding the representation $\mathbf{h}$ built by the network. The loss is minimized using the ADAM optimizer (Kingma & Ba, 2014), with a learning rate linearly decreasing from $10^{-3}$ to $10^{-5}$ in $10^6$ epochs, and a mini-batch size of $100$ sensorimotor transitions. The optimization is stopped after $2 \times 10^6$ epochs, or when the training loss falls under a threshold set to $10^{-8}$ on a mini-batch.

*Training data:* Training data are generated by having the simulated agent explore its environment and collect sensorimotor transitions $(\mathbf{m}_t, \mathbf{s}_t) \rightarrow (\mathbf{m}_{t+1}, \mathbf{s}_{t+1})$. A total of $3 \times 10^6$ transitions are collected for each simulation by randomly sampling the motor space (uniform distribution) and getting the corresponding sensory inputs generated by the environment. If a drawn motor state corresponds to an impossible spatial configuration of the sensor (sensor lying outside of the environment, or inside an object) no sensory input is received and the sensorimotor transition is discarded. Depending on the type of exploration, the environment also moves during the data collection, instantaneously changing its position relative to the agent. The environment can randomly translate (uniform distribution) with a maximal amplitude equal to its size (computed relatively to the environment's initial position at the start of the simulation):

- In the MTM case, the environment translates between the collection of $(\mathbf{m}_t, \mathbf{s}_t)$ and $(\mathbf{m}_{t+1}, \mathbf{s}_{t+1})$. This ensures that the sensorimotor transitions do not fulfill condition I, and a fortiori condition II.
- In the MM case, the environment never translates and keeps its initial position during the collection of all sensorimotor transitions. This ensures that the sensorimotor transitions fulfill condition I, but not condition II.
- In the MMT case, the environment translates each time 100 transitions $(\mathbf{m}_t, \mathbf{s}_t) \rightarrow (\mathbf{m}_{t+1}, \mathbf{s}_{t+1})$ have been explored. Note that some motor states might correspond to impossible spatial configurations. The number of effectively collected sensorimotor transitions might thus vary from 0 to 100 for each translation of the environment[1]. This data collection ensures that the sensorimotor transitions fulfill conditions I and II.

For each training epoch, 100 quadruplets $(\mathbf{m}_t, \mathbf{s}_t, \mathbf{m}_{t+1}, \mathbf{s}_{t+1})$ generated this way are randomly drawn to form the mini-batch. Before being fed to the network, the data is normalized such that each component spans $[-0.8, 0.8]$ over the whole dataset.

*Remarks:* The overall neural network architecture and training procedure have been kept simple. No particular heuristics have been added to improve convergence, generalization, or any other property of the network such as sparsity. Similarly, the architecture's meta-parameters have not been optimized beyond simply checking that the network was expressive enough to learn the expected

---

[1] In practice, this indirectly simulates an environment that has some non-zero probability to move after each sensorimotor transition.

mappings. Moreover, the network's generalization capacity has not been evaluated. This is because the prime goal of this work is not to optimize a neural network to efficiently solve a task, but rather to study how spatial invariants get naturally captured as a byproduct of sensorimotor prediction, without the need for additional priors. We even observed, in additional experiments not reported in this paper, that spatial topology and metric regularity get captured even when the sensory prediction is of very low accuracy, due for instance to very ambiguous sensorimotor experiences.

The code to run the simulations, train the neural network, and analyze the motor representation will be accessible at [*anonymous url*].

## B    DETAILED ANALYSIS OF THE RESULTS

In addition to the qualitative result description proposed in Sec. 5, we develop below a more thorough analysis of each learning curve. As a reminder, each simulation is run 10 times, with all random parameters drawn independently on each trial. The evolution of the mean and standard deviation of the loss, $Q_p$, and $Q_h$ is displayed in Fig. 2. The measures $Q_p$ and $Q_h$ are computed using a fixed regular sampling of the motor space in order to ensure consistent comparisons. Additionally, Fig. 3 shows the final representation of the same fixed motor sampling, for one randomly selected trial of each simulation. The corresponding ground truth positions, as well as the affine compensations between the representation space and the position space are also displayed.

### B.1    DISCRETE WORLD

We first analyze the impact of the exploration on the representation in the discrete world setup.

*MTM exploration*: We can see in Fig. 2 that the loss rapidly drops, due to the network finding an adequate average value and scale for $\tilde{\mathbf{s}}_{t+1}$. It then stagnates around a high value ($\approx 0.7$). This shows that the network struggles to learn an accurate prediction. This is expected as the random transitions of the environment between sensorimotor pairs $(\mathbf{m}_t, \mathbf{s}_t)$ in the database make the sensory experiences inconsistent. The dissimilarity measure $Q_p$ stays high ($\approx 0.13$) during the learning and slightly oscillates. This oscillation is reduced after $10^{-6}$ epochs due to the learning rate reaching its minimal value. The dissimilarity measure $Q_h$ also stays high ($> 0.12$) during the learning. It however goes through an initial phase of significantly greater values ($\approx 0.2$) before progressively decreasing to its base value ($\approx 0.12$). This is better illustrated in Fig. 4 where the training has been pursued for $4 \times 10^6$ epochs. The same figure also shows a comparison with a network using ReLu units (Glorot et al., 2011) in place of the SeLu units. It suggests that the initial overshooting of $Q_h$ is to be attributed to the SeLu units initialization and self-normalizing dynamic, and not to a particular property of the sensorimotor experience. (Note also that no qualitative difference was observed in the structure of the representation built by the network when using ReLu units).
In agreement with these observations, Fig. 3 shows that the motor representation $\mathbf{h}$ does display an arbitrary structure compared to the regular one of the ground-truth position $\mathbf{p}$. This is expected as the sensorimotor pair $(\mathbf{m}_t, \mathbf{s}_t)$ is not informative to predict $\mathbf{s}_{t+1}$, due to the environment moving between $t$ and $t + 1$. As a consequence, the network learns to output an average sensory state $\tilde{\mathbf{s}}_{t+1}$ which statistically minimizes prediction error while not relying on $\mathbf{m}_t$, $\mathbf{s}_t$, or $\mathbf{m}_{t+1}$. The network thus builds an arbitrary representation $\mathbf{h}$ which is disregarded by the predictive module.

*MM exploration*: The experience of consistent sensorimotor transitions greatly impacts the loss and dissimilarity measures $Q_p$ and $Q_h$. The environment being static and unambiguous by construction, the predictive task is now trivial. The loss rapidly drops to very small values of the order of $10^{-8}$. Because of the stopping criterion set to a loss inferior to $10^{-8}$, the training can end before the maximal number of epochs set to $2 \times 10^6$ epochs. In such a case the network is frozen for the remaining number of epochs, such that mean and standard values can be computed over the whole training time. In practice, the last of the 10 independent trials reached the $10^{-8}$ threshold after $7.6 \times 10^5$ epochs. Measures $Q_p$ and $Q_h$ undergo a rapid decrease and stabilization. The dissimilarity $Q_p$ stabilizes around values which are significantly lower than in the MTM case ($\approx 0.07$), while $Q_h$ stabilizes around values which are similar to the MTM case ($\approx 0.1$). These results correlate with the representation displayed in Fig. 3. We can see that the affine projection of $\mathbf{h}$ in the position space gives rise to a point cloud with the same topology as the ground truth position $\mathbf{p}$. In particular, all redundant motor states $\{\mathbf{m} = [m_1, m_2, m_3], \forall m_3\}$ are projected onto the same point $\mathbf{h}$. In the

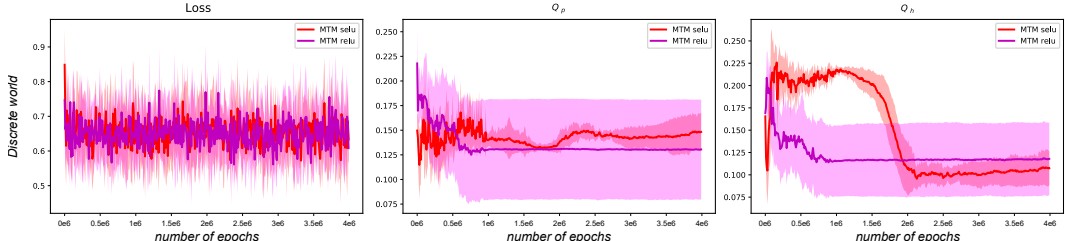

Figure 4: Evolution of the loss and the dissimilarity measures $Q_p$ and $Q_h$ during a longer training session for the discrete world and the MTM type of exploration. Two networks using SeLu units and ReLu units are compared. The displayed means and standard deviations are computed over 10 independent trials. (Best seen in color)

3D representational space we can however see that the representations $\mathbf{h}$ of the motor sampling do not lie on a flat 2D manifold. They are instead spread in the whole space. Indeed, the constraints induced by the topological invariants drive the motor representation to copy the topology of the sensory manifold. Nevertheless these topological constraints can be respected by a curved 2D manifold in the representational space. During training, the representation can thus converge to an infinity of such solutions, of which a flat manifold is only a special case.

This analysis also leads us to conclude that $Q_p$ and $Q_h$ are not ideal criteria to compare the topologies of $\mathbf{h}$ and $\mathbf{p}$. Indeed they implicitly assume a linear mapping between the two manifolds, while it is not the case in practice. The dissimilarity measures $Q_p$ and $Q_h$ thus act as an upper bound of a true topology dissimilarity, which would produce lower values by taking into account the non-linearity between the two manifolds. Performing such a non-linear comparison would however require a significantly more complex comparison of the two point clouds. Although not ideal, $Q_p$ and $Q_h$ are still sufficient to compare the impact of the different types of exploration on the topology of $\mathbf{h}$. Finally, we can also notice that $Q_p$ appears to be a more adequate criterion than $Q_h$ in our study. Contrary to $Q_p$, $Q_h$ fails to capture the structural difference between the MTM and MM cases, due to the fact that $\mathbf{h}$ can be spread in the 3D representational space (see Fig. 2).

*MMT exploration*: Adding translations of the environment between sensorimotor transitions greatly impacts the loss and dissimilarity measures. The loss rapidly drops to very small values ($\approx 3 \times 10^{-6}$) although more slowly than in the MM case. Despite the predictive task being made more complex by the environment moving, the network is thus able to produce accurate sensory predictions. The dissimilarity measures $Q_p$ and $Q_h$ also drop to very small values ($\approx 2 \times 10^{-3}$, and $\approx 3 \times 10^{-3}$ respectively). This indicates that there exists a simple affine transformation between the representation $\mathbf{h}$ and the ground-truth position $\mathbf{p}$; they should have a similar topological and metric structure. This is confirmed in Fig. 3, where we can see that both $\mathbf{h}$ and $\mathbf{p}$ align perfectly after affine compensation in the position space. Therefore the agent has been able to capture both the topology and the metric regularity invariants in its representation. As a result, $\mathbf{h}$ is a reliable proxy for the sensor position $\mathbf{p}$ in the external Euclidean space.

We can see in Fig. 3 that the representation $\mathbf{h}$ even approximates a flat 2D manifold in the 3D representational space. This is not however the case in all trials, as shown in Fig. 5. Indeed, in the overall network, the representation $\mathbf{h}$ is fed to a fully connected layer. Each neuron in this layer performs a linear projection of $\mathbf{h}$ before passing it through its activation function. As a consequence, the network has two equivalent options to respect the sensory invariants induced by the MMT exploration: i) flatten the 2D motor representation manifold in the 3D representational space such that all the metric equalities are respected, or ii) adapt the weights of the connection to the fully connected layer such that all neurons perform projections which take into account only two dimensions in the representational space. In both cases, the input received by the predictive module Net$_{\text{enc}}$ would be equivalent. Depending on the trial, it seems that the network converges arbitrarily to one of those two solutions, or rather a mix of the two, often displaying a slight curvature of the 2D manifold in the representational space. This explains the slightly greater standard deviation of $Q_h$ compared to $Q_p$ observed in Fig. 2. Note that even when the network converges to a curved manifold, the projective nature of the second option ensures that the affine projection of $\mathbf{h}$ in the space of $\mathbf{p}$ preserves the metric constraints induced by the sensory invariants. This explains why the seemingly spread cloud

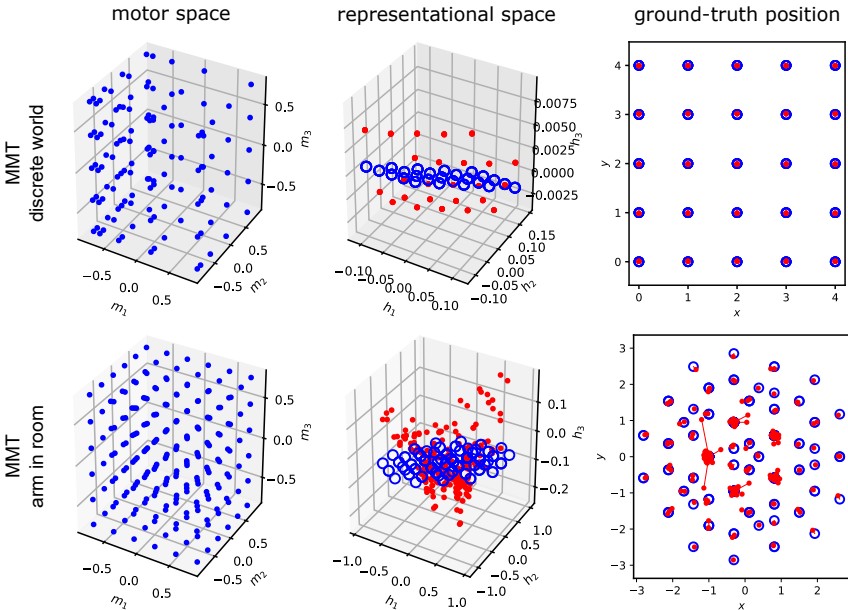

Figure 5: Visualization of the normalized regular motor sampling **m** (blue dots), its encoding **h** in the representational space (red dots), and the corresponding ground-truth position **h** (blue circles) for the MMT type of exploration, for different trials of the discrete world (a) and arm in a room (b) setups then in Fig. 3. Both the representations and the positions are also projected into each other's space via an optimized affine transformation. Lines have been added in the ground-truth position space to visualize the distances between the projections of **h** and their ground-truth counterparts **p**. (Best seen in color)

of points **h** observed in Fig. 5 corresponds to a perfectly regular grid when projected in the space of **p** via affine transformation.

## B.2  ARM IN A ROOM

We now analyze the impact of the exploration on the representation in the more complex arm in a room setup.

*MTM exploration*: Like in the discrete world setup, the sensorimotor predictive task is made difficult by the translation of the environment occurring between sensorimotor pairs $(\mathbf{m}_t, \mathbf{s}_t)$ and $(\mathbf{m}_{t+1}, \mathbf{s}_{t+1})$. As a consequence, the loss stagnates around a high value ($\approx$ 1.0), after a quick drop due to the network finding an adequate average value and scale for $\tilde{\mathbf{s}}_{t+1}$. Additionally, both $Q_p$ and $Q_h$ progressively decrease during the learning, before reaching a plateau at relatively high values ($\approx$ 0.12 each). This progressive decrease was not observed in the discrete world setup. We hypothesize it is due to a border effect related to the environment having walls. Indeed the world is not a torus in this setup. As a consequence, the motor states corresponding, for instance, to the sensor being to the left of the agent's base rarely observe the sensory states associated with positions of the sensor on the far right of the environment. Similarly, different positions of the sensor statistically observe slightly different sensory distributions. This drives the network to capture an approximation of the topology of **p** where, in particular, redundant motor states **m** tend to be associated with the same sensory distribution and thus tend to be represented by the same **h**. This explains why the dissimilarity between **h** and **p** measured by $Q_p$ and $Q_h$ decreases during training. Note however that this mechanism is not sufficient to capture the true topology of the sensor position as it is based on the statistics of the sensory states, and not on their absolute values (different sensorimotor mappings could lead to similar sensory statistics without necessarily inducing the topological invariants described in Sec. 3). The representation displayed in Fig. 3 indeed shows that the representation built by the network seems less arbitrary than in the discrete world setup. Motor states corresponding to close sensor positions tend to appear closer in the representation than in the discrete world setup. Yet the **h** still does not capture the true topology of the position **p**.

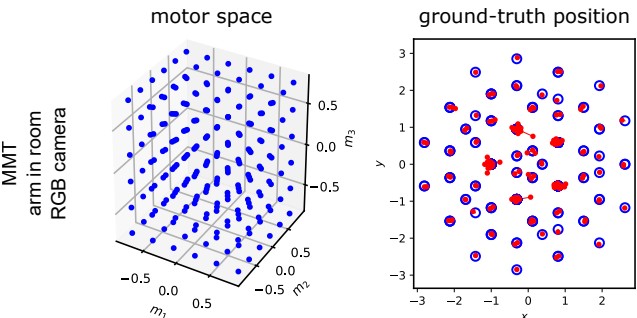

Figure 6: Visualization of the (normalized) regular motor space sampling, the corresponding ground-truth positions (blue circles), and the affine transformation of the representation into the ground-truth space of positions (red dots) for one trial of the arm in a room simulation with a RGB camera of 16 pixels, a representational space of dimension $dim_H = 25$ and a MMT type of exploration. (Best seen in color)

*MM exploration*: In the static environment, the loss quickly drops to small values ($\approx 0.07$). This indicates that the sensorimotor predictive task is relatively easy, due to the environment being static, and despite the sensorimotor interaction being more complex that in the discrete world setup. The measures $Q_p$ and $Q_h$ also rapidly decrease, before reaching a plateau ($\approx 0.09$, and $\approx 0.12$ respectively). This seems to indicate that $\mathbf{h}$ is more structurally similar to $\mathbf{p}$ than in the MTM case. This is confirmed in Fig. 3, and in particular in the position space where the representation are projected via an affine transformation. Despite some outliers, we can see that $\mathbf{h}$ topologically organizes as the star-shaped grid of ground-truth positions $\mathbf{p}$ generated by the regular motor sampling. In particular, the multiple redundant motor states $\mathbf{m}$ leading to the same external sensor position $\mathbf{p}$ are clustered together (see for instance the inner corners of the star). In the representation space, the states $\mathbf{h}$ appear spread in the 3D space, for the same reason as described for the discrete world setup. Finally, these results confirm that $Q_h$ is not the best measure to analyze the structure of the representation. In Fig. 2, it fails to capture the structural difference between the MTM and MM scenarios, whereas $Q_p$ does.

*MMT exploration*: With consistent sensorimotor transitions in a moving environment, the loss rapidly decreases and reaches a plateau around a small value ($\approx 0.25$). The network is thus able to learn a relatively good sensorimotor predictive model. The residual error is however greater than in the discrete world setup, in part because the sensory experience in this environment can be ambiguous. For instance, when sensing only the right wall, the environment could be in many different positions, and the agent thus cannot predict the future sensory state with certainty. More importantly, $Q_p$ and $Q_h$ also drop to very small values ($\approx 0.014$). This indicates a high structural similarity between $\mathbf{h}$ and $\mathbf{p}$. This is confirmed in the ground-truth position space of Fig. 3. After affine projection, we can see that the representation $\mathbf{h}$ almost perfectly match the star-shaped grid of positions $\mathbf{p}$ induced by the regular motor sampling. Therefore the agent has been able to capture both the topology and the metric regularity invariants in its representation. As a result, $\mathbf{h}$ is once again a reliable proxy for the sensor position $\mathbf{p}$ in the external Euclidean space. Note that in this trial the representation $\mathbf{h}$ once again approximates a flat 2D manifold in the 3D representational space. As shown in Fig. 5, this is not necessarily the case in all trials, for the same reason as discussed in the discrete world setup.

### B.3 ADDITIONAL EXPERIMENT

We proposed a last additional experiment in order to assess the robustness of the approach to more complex sensory inputs and a representational space of higher dimension. The arm is now equipped with a 1D RGB camera (in the 2D room) with a resolution of 16 pixels. Additionally $dim_H$ is set to 25 instead of 3. The resulting learning curves are displayed in black in Fig. 2, for the MMT exploration case. Like with the initial arm in a room setup, the loss progressively decreases and reaches a plateau ($\approx 2.1$). The predictive loss thus exhibits a similar behavior, despite its overall greater magnitude, due to the MSE being computed in a sensory space of dimension 48 instead of 10. Likewise the evolution of the dissimilarities $Q_p$ and $Q_h$ is qualitatively identical to the case of

the simpler arm, despite the largely greater dimension of the representational space. The measures even converge to identical values. This seems to indicate that the network built a representation $\mathbf{h}$ which is structurally similar to the ground truth position $\mathbf{p}$. This in confirmed in Fig. 6 where the affine projection of $\mathbf{h}$ in the the ground-truth position space matches $\mathbf{p}$ (the representational space is now of too high dimension to be straight-forwardly visualized). The RGB camera experiment was also run with $dim_H = 3$, and generated qualitatively similar results. The approach thus seems insensitive to the complexity of the sensory input, and to the dimension of the representational space.

