# OpenReview forum: "Unsupervised Emergence of Spatial Structure from Sensorimotor Prediction"
_ICLR.cc/2019/Conference_

### Official Review · AnonReviewer2 · 2018-11-02
**Unconventional and interesting approach to unsupervised learning of sensorimotor perception**

**Rating:** 7
**Confidence:** 3

**Review:**

This very interesting paper is based on the sensorimotor contingency theory, which grounds the perception of the agent (motor perception and sensory perception) in the capacity to learn to predict future sensory experience and to build a compact internal representation of proprioception and motor states. They show that through active experience (exploration) of an environment, an agent can encode its motor state in a way that captures both the topology and the relative distances. The authors show that in the case of an environment consistent with sensorimotor transitions and with some changes in the environment that are not inconsistent with the sensorimotor transitions (resets?), the network can learn the representation h of motor that project to the actual position of the agent in the environment (albeit in very contrived 2D toy tasks).

The model does not rely on RL at all; rather, it uses the representation of motor states (proprioception) to make predictions about the sensory observations of the environment. If such an agent were to act, I presume that there would be a search procedure across motor states to make the agent reach a desired state. The paper merits publication at ICLR provided very extensive revisions are made, and I am listing here the improvements to be made.

It is cool to cite Kant, Poincaré and Nicod, whose philosophical work underlies subsequent work on representing space and sensory experience. When citing Kant, please cite the primary source, not the 1998 re-edition.

There are missing references to Wayne et al (2018) "Unsupervised Predictive Memory in a Goal-Directed Agent" arXiv:1803.10760 and Ha & Schmidhuber "World Models" arXiv:1803.10122, where an agent is shown to build a representation of the world that can be decoded into spatial position and even a map of obstacles, on previously unseen environments, using only prediction of images, rewards and actions. Please include them in your work as they considerably change the narrative of section 2 (related work). Essentially, while the claims of the paper are interesting and relevant for the representation learning community, similar work has already been done, at much larger scale, from visual observations, using RL and self-supervised learning.

Section 2 is also somewhat overly critical of previous work: in (Banino et al, 2018; Cueva & Wei, 2018) "rely[ing] on extraneous spatial supervision signals [do not] counter any claim of autonomy", first because these signals can come from sensory perception (e.g., smell) and also because the agent is still autonomous at test time. Similarly, the depth prediction task in (Mirowski et al, 2017) is rather intuitive (stereo-vision).

Part 3 is difficult to parse: it would help to use the word proprioception (or explain why it is inappropriate) when talking about motor states, and exteroception (sensing the environment). I understood the first assumption, which is local continuity in sensory and motor space as well as the ambiguity of redundant motor systems that can generate the same sensory states, but not the second assumption. From what I understand, there are two invariants in building the model of proprioception: invariance to the topology of the environment and to the distances between objects, but then this is hard to reconcile with the setup of (in)consistent transitions in a moving environment and consistent transitions in a static environment. Please rewrite this section in a way that is easier to parse for people who know state-space models and RL for navigation and grasping (who may be your audience). Moreover, all the references point to a single work, which suggests that it is a very peculiar way of approaching a much more general problem of sensorimotor prediction, and therefore begs for a clear and simple explanation.

The architecture of the model is interesting: typically deep RL papers encode the sensory observations s into a hidden representation h, to take actions and produce a motor state m. Here, the current and future motor states m_t and m_{t+1} are embedded into h_t and h_{t+1} using a siamese MLP and used, in combination with the current sensory observation s_t, to make a prediction of s_{t+1}. This is somewhat related to learning the dynamics in model-based RL; please look into and cite Pathak et al (2017) "Curiosity-driven exploration by self-supervised prediction", ICML and other work on intrinsic curiosity.

The experiments are in very simplistic 2D grid world environments, but it makes the analysis and understanding of the 3D representations h much more simpler to follow. On the other hand, the discrete world task is very contrived (especially the weird mapping from m to h and from p to s) and hard to relate to existing work. One difficult problem that is solved by (Banino et al, 2018) is that of path integration in 2D from egocentric velocity. Could the authors present results on such a nonlinear case?

Revision: Score updated from 6 to 7.

---

> ### Author Response · Authors · 2018-11-22
> **Reply to reviewer 2 - part 2**
>
> We agree with you that our work somehow relates to a large field of work about sensorimotor mapping learning, action-state mapping learning, or even intrinsic curiosity (usually used in model-based RL). We added some of them in the Related work section, but couldn’t include all of them, due to the limited space.
>
> We agree that the discrete world setup appears very artificial. It has been purposefully designed to exhibit all the properties that our approach requires to perfectly illustrate the emergence of spatial structure. The small grid world is easy to explore, and analyze. It is a torus in order to avoid the border effect that is present in the arm in a room simulation (see Appendix B.2 – MTM exploration). The sensory experience continuously changes with respect to the position of the sensor in the world. The sensory input is sufficiently complex (4D) to avoid any sensory ambiguity between different positions of the sensor. And the motor-to-position mapping is non-linear and redundant, so that we can show how the metric from the motor space gets modified in order to build the representation h.
> Although it is obviously non-realistic, this contrived setup is useful to illustrate the core ideas of the approach. Its purpose is the same as initial toy-examples that can be seen in other papers, like for instance in (Thomas et al., 2017).
>
> Finally, we are not sure to understand what you mean by “such a nonlinear case”, when referring to (Banino et al, 2018). Unless we misunderstood your question, the setups we consider are already non-linear. The motor-to-position mapping, as well as the position-to-sensation mapping are both non-linear in the discrete world and arm in a room setups.
> Regarding path integration, this is a question we have not investigated explicitly in this paper. Some work has however been done about this specific problem in (Ortiz et al., 2018). They show that a recurrent network can learn to integrate sequences of egocentric velocity commands to estimate the agent’s position, based only on sensorimotor prediction.
>
> We significantly reworked the manuscript to take into account all reviewers’ feedback. In particular, we clarified the formalism and improved the wording, which was often too verbose or vague. We hope that the paper is now more compelling and better delivers its message, which we believe to be of interest for the Representation Learning community.

---

> ### Author Response · Authors · 2018-11-22
> **Reply to reviewer 2 - part 1**
>
> We would like to thank you for your constructive feedback. We have revised the manuscript to improve the problem description, the mathematical formalization, and the simulations description. We also greatly modified the Related work section to take your suggestions into account.
> We answer your additional questions in detail below.
>
> As you suggested, the final motor representation could be used to perform reaching tasks. This was partially illustrated in (Laflaquière et al., 2018), where an optimal trajectory to the target is determined by applying the Dijkstra algorithm. One could also train a network to generate motor trajectories, based on the spatial representation. This is not something we present in this paper, which is focused on the emergence of the spatial representation itself.
>
> We modified the bibliography to refer to Kant’s original work from 1781. Your suggestion regarding other approaches based on predictive models of the worlds, like (Ha & Schmidhuber, 2018) and (Wayne et al., 2018), was very pertinent, and we added them to the Related work. We also highlight how our approach differs from theirs. The representations they learn are tailored to the agent’s sensorimotor apparatus (like us), but also to the content of the environment. As such, the representations they build are not only spatial. They also do not explicitly represent the egocentric position of the agent’s sensor, but rather the allocentric position of the agent in the world (which needs to be decoded using a supervised approach). They also cast the problem in a Reinforcement Learning setup in which the reward can play a role in shaping the representations.
>
> We reworked the Related work section so that it appears less critical, which wasn’t our original intention. We do not consider that the various works we mention and that use priors are irrelevant. However, we differ from them in that we try to show that sensorimotor prediction might be sufficient to capture spatial structure. Nevertheless, we still think that the spatial supervision signal used in (Banino et al, 2018; Cueva & Wei, 2018) is a strong limitation, as it couldn’t be simply inferred from raw sensory inputs, especially in complex environment. Finally, the auxiliary tasks proposed in (Mirowski et al, 2017) are indeed intuitive, and make sense when one wants to solve spatial tasks. We just emphasize that they are extraneous priors, designed to facilitate the emergence of spatial content in the representations, while we try to avoid any such prior.
>
> We significantly reworked the Problem setup section to make it clearer. The new revised version points out that the motor state m can be seen as a direct posture, or as a proprioceptive reading in case the agent is not controlled in position.
> Both the topological and metric invariants induced by spatial exploration are now introduced in a more formal way, using equations that the reader can more easily parse and refer to. The relation between these invariants and the three types of exploration has also been made clearer by defining two conditions that the sensorimotor experience can have. We hope that this new presentation of the problem formalization is more compelling.

---

### Official Review · AnonReviewer1 · 2018-11-03
**Interesting topic, contribution questionable, better write-up required**

**Rating:** 6
**Confidence:** 3

**Review:**

The paper proposes that agents can extract the spatial structure of the world if consistently explored solely from sensorimotor predictions.

The topic of the paper is relevant and fits the conference. However, I have doubts about the generality of the claims made.

Strong aspects:
Abstract makes the reader curious and excited about the content
Interesting and important topic for representation learning.

Weak aspects:
The paper is written in a way that it is not easy to follow.
Important details are in the Appendix and some more formal description would be helpful. A clearer presentation of the three different exploration types for instance would help the reader.
The paper is not using standard notation and deviates from the POMDP description for not clear reason.
It is hard to compare and the results are in a very restricted setup.
No baseline or comparison to other methods are given.
Why does space emerge in the motor-state (h) not the sensor state?
Not enough details given to reproduce the paper. It is nothing mentioned about code to be made public.

Discussion:
The argument of the paper is that it is sufficient to do sensor prediction when the world is consistently explored and changing in a consistent manner. There is no need for additional losses or the like. However, I am not convinced about the generality of this statement. What if the world is much richer and the agent and the environment undergo the typical transformations. Rotation and movement in 3D space + non-linear distortions by sensors (e.g. a camera). I am not seeing why a representation of space should emerge without any pressure for minimality or the like. The paper also assumes that the sensor is moved by the motor command in space directly and not, for instance, as a double integrator equations where the motor command is a force that accelerates a body.
In the current form, what it really says, is that the network figures out the actual effect of the motor command in moving the sensor around. The sensor is moved in a 2D space by the actions such that the network recovers this 2D manifold. And this can only happen if it actually observes movements and the dataset is rich enough. This distinguishes the 3 different exploration settings.

Details:
Sec 2:  Prior work: Which priors in Jonschkowski and Brock 2015 do you mean that are specific to the emergence of space? Slowness?
Sec 3: what do you really mean by motor state? In MDPs there is the notion of action that effects the state of the system. There is not really a state of the motor system? Do you mean proprioceptive sensors or actually the motor command/action.
The definition s = \phi_e(m) is kind of nonstandard as one would expect a dependency on s_{t-1} which is in your notation hidden in \epsilon.
The logic is not so straight forward to me. Isn't the logic such that: Given a rigid metric space: when the agent moves around the same type of movements lead to the same kind of transformations independently of there the agent is located?
I believe this section could be streamlined and illustrated by some examples to drive the message home. Also, making a clear and potentially more explicit statement of these invariances (e.g. by an equation) and why they will be revealed by learning predictions.

Sec 4.2:
Improve the description of the settings. In the first setting (MTM) subsequent sensor/motor values are independent right? So it is the same as having randomly selected isolated data points from a normal interaction.

MMT setting: you write ... which can translate randomly after each transition.  This is almost the same as in MTM where you write ...translates randomly between t and t+1
In the Appendix you write that in MMT that environment moves only every 100 steps?

BTW:
Is the env-movement a smooth movement or a jump?

Fig 2: why does the green curve end so early? Is it because of your stopping criteria for training. I would like to see the same training time for all settings.

Fig 3 and text: Should one not expect a torus in (a)? The world is not a square but a torus, as you have cyclic boundary conditions. I am surprised to not see this in the plots. The current result somehow violates the smoothness because there is a big jump between the boundaries although in environment there is none.

Sec 6: par 2: ...these invariants represents for the predictive model?

Out of my curiosity:
you write that ... casts some doubts on purely passive and observational approaches....
In which sense did the actions help here? Do you mean that the agent needs to know its own actions right?
So when it is to be done from a video (just sensor information) than the actions would need to be inferred first?

Typos:
p1 par2: approache
p8 last par: could be merge

updated score after authors revision

---

> ### Author Response · Authors · 2018-11-22
> **Reply to reviewer 1 - part 4**
>
> Fig 2: We modified Figure 2 to see the result of the learning up to 2x10^6 iterations, for the three types of exploration.
>
> Fig 3 and text: The environment explored by the agent in the discrete world setup is indeed a torus. However, the spatial representation built by the agent corresponds to the position of its sensor in an egocentric frame of reference. As a result, it appears as a flat working space with outer limits (5x5 grid in this case); a bit like an ant on a sphere would describe its surrounding as a flat disk.
> The egocentric nature of the representation has been made clearer in the revised manuscript.
>
> - About passive approaches:
> The importance of the motor components and of the active exploration for the emergence of spatial knowledge is indeed a line of argument going back to (Poincaré, 1895). Unless some specific a priori knowledge can be hard-coded in the system, the only way for a naive agent to discover that spatial displacements exist is to move. The agent can discover this way that some sensory changes it produces, and some sensory changes that the environment produces have particular properties that other don’t have: they can compensate each other. This “compensability” property is, according to Poincaré, the foundation of our notion of space. This specific property of displacements is not explicitly presented in this work, although it underlies the ability of the agent to capture metric invariants. If you are curious about the nature of this phenomenon, the compensability of displacements is discussed at length in (Poincaré, 1895), (Philipona et al., 2013), and (Laflaquière et al., 2018).
> Following this reasoning, a naive agent without access to its own motor commands or proprioception shouldn’t be able to discover the structure of space. As you suggested, in case the agent has only access to the sensory flow, actions would need to be inferred first. Nonetheless, this appears to be an impossible endeavor without prior knowledge, as a naive agent wouldn’t be able to distinguish sensory changes coming from the environment from sensory changes due to its own actions. As such, this line of argument also loops back to some causal inference ideas, put forward by J.Pearl.
>
> Typos: Thank you for pointing to several typos, which have been corrected in the revised manuscript.
>
> We significantly reworked the manuscript to take into account all reviewers’ feedback. In particular, we clarified the formalism and improved the wording, which was often too verbose or vague. We hope that the paper is now more compelling and better delivers its message, which we believe to be of interest for the Representation Learning community.

---

> > ### Comment · AnonReviewer1 · 2018-11-26
> > **Improved, but still issues**
> >
> > I am acknowledging the detailed answers to my concerns and the changes made to the manuscript.
> >
> > The new mathematical formulation is helpful, although has some small problems, see below.
> > I have, unfortunately, still concerns:
> > 1) I still don't think your experiments and theory justifies the claims that there are not priors needed to recover space but just prediction. It is true in your artificial examples, but I am pretty certain it will break down as soon as you consider more general cases.
> > Your example: "[...] Imagine the agent has been through multiple environmental states, and discovered that m_i and m_j have always generated identical sensory states. By representing m_i and m_j the same way, it gains the ability to extrapolate to new situations.[...]" is also not convincing.
> > A pure correlation-based learning would not force m_i and m_j to be represented in the same way. The method would produce the same result for m_i and m_j on the observed domain but can be differently represented. Outside the observed data (on extrapolation) there is absolutely no guarantee what the predictor will do. If you would require a minimal representation/network structure, then you could enforce m_i and m_j to be treated equally.
> > 2) Discrete world and Fig 3: I really don't understand the point of egocentric vs allocentric view for the torus setting. The topology does not change, a torus remains a torus. The representation does not preserve the smoothness of the world. A smooth transition over the cyclic boundary condition (which should be transparent to the agent) will cause a jump in the representation. How can that be a correct representation of the space the agent lives in?
> >
> > Details:
> > - Eq (1) and (2): I am pretty sure you need a constant in front of \mu on either side (Lipschitz - constant)
> > - I suspect that \vec{p_i p_j} means the vector from p_i to p_j. Why not writing p_j - p_i then?

---

> > > ### Author Response · Authors · 2018-11-26
> > > **Additional reply to reviewer 1 - part 2**
> > >
> > > Details:
> > > - We understand your suggestion regarding the Lipschitz-constant. However, the conditions described by Equations (1) and (2) are not about Lipschitz continuity, which would be a global property of the sensorimotor mapping, but more about local properties, neighborhoods and differential topology. \mu can be seen as small value which tends towards 0.
> > > - \vec{p_i p_j} indeed means (p_j – p_i). We tried both notations when writing the equations and the \vec{.} option appeared to be the more readable, especially in equation (2).

---

> > > > ### Comment · AnonReviewer1 · 2018-11-26
> > > > **Reply to Additional reply to reviewer 1 - part 2**
> > > >
> > > > - I understand that the formula is a local definition, but even in the limit \mu -> 0,
> > > >  you are assuming that delta-m induces a larger deviation in p than in s, right?

---

> > > > > ### Author Response · Authors · 2018-11-26
> > > > > **Short reply to reviewer 1**
> > > > >
> > > > > Unless we misunderstand your question, the answer is no.
> > > > >
> > > > > Let's focus on Equation (1), which is simpler to parse. We assume that by "delta-m" you mean (m_j - m_i). Then Equation (1) means that for any such (m_j - m_i) of any amplitude, if the corresponding displacement (p_j - p_i) tends towards 0, then the sensory changes (s_j - s_j) also tends towards 0. Regardless of the amplitude of the motor change (which can be great), both the deviation in p and s are thus small.
> > > > >
> > > > > The same reasoning applies to Equation (2), except that we now consider differences between vectors, instead of the amplitude of vectors themselves.

---

> > > ### Author Response · Authors · 2018-11-26
> > > **Additional reply to reviewer 1 - part 1**
> > >
> > > Thank you for reading the new version of the manuscript, and expressing your remaining concerns.
> > > 1)	We’ve been careful not to make the strong claim that priors *are not* needed to discover spatial properties. In the Abstract, Introduction, and Conclusion, where the idea is discussed, we purposefully use the subjunctive mood, and verbs like “suggest” to emphasize that we discuss a possibility. This possibility has been evaluated and validated on simple simulations first. This is, in our opinion, a typical procedure when evaluating new ideas, like in (Banino et al., 2018), (Cueva et al., 2018), (Jonschkowski and Brock, 2015), (Philipona et al, 2003), (Terekhov and O’Regan, 2016), (Thomas et al., 2017), etc.
> > > Like in those works, the results are not a definitive proof that the proposed mechanisms straightforwardly scale up to significantly more complex cases. In our case, the Conclusion section, in particular, lists some current limitations that need to be overcome to extend the approach. Yet, we think that our simulations and results demonstrate, in simple examples, that sensorimotor prediction can indeed lead to the capture of spatially-induced sensorimotor invariants. We thus think that this mechanism should be taken into account when addressing the problem of learning spatial representations.
> > > Of course, if you see a counter-argument to challenge the possibility of such a mechanism, please inform us, so that we can revise the manuscript or add it to the limitations listed in the Conclusion section.
> > >
> > > Regarding the examples about m_i and m_j having the same representation. We agree with you that, in theory, the sensorimotor predictive mapping should be able to generate the same predictions, even if m_i and m_j are represented differently. We explicitly dedicate the end of the first Conclusion paragraph to this idea. We point out that, although it is possible in theory, in practice the network rather converges to a representation which captures the sensorimotor invariants, and represents the same way m_i and m_j. This was the case in all simulations we tested (many of them are not presented in the paper). We add in the conclusion that understanding why this particular configuration acts like such an attractor for the motor encoding is an interesting question to investigate.
> > > Regarding the “extrapolation”, our previous reply might have been misleading. Our argument was twofold. a) During training, representing m_i and m_j differently seems to induce prediction errors. Intuitively, this can be understood as having to learn the structure of the sensory manifold twice (separately for both m_i and m_j), which leads to twice the amount of error in the prediction. Whereas when m_i and m_j are represented by the same h, the predictive module only has to learn the structure of the sensory manifold once, which generates less errors overall (the optimization of the predictive mapping for m_i also affects m_j). We think this is this difference that drives the network to represent m_i and m_j with the same h, although the precise mechanism needs to be more formally described. The term “extrapolation” was not the best term to describe this idea. b) The term “extrapolation” has more sense when considering a full cognitive architecture, at a more fundamental level. This is not done in this paper, and this idea was thus only mentioned in our previous reply. When considering a full cognitive system, with additional modules like long term and working memories, the similar representation of m_i and m_j offers significant benefits. In that context, whenever the agent experiences a new sensory input s_new with the motor state m_i, it can naturally extrapolate (never experienced before) that m_j would *also* generate s_new. This however falls out of the scope of this work, and would require additional modules to which the motor encoding h would be an input.
> > >
> > > 2)	You are right that a torus remains a torus. The point about egocentric vs allocentric is the following: the spatial representation built by the agent is not a representation of the environment (all positions the agent could be in the environment), but a representation of its working space (positions its sensor can reach relative to its base). Because the agent’s working space is smaller than the environment (5x5 sub-grid in a 10x10 torus grid), the egocentric spatial representation it ends up building is simply a 5x5 grid. Because this working space is only a sub-part of the whole torus environment, it doesn’t itself act as a torus. Said otherwise, when the agent moves its sensor in the working space, it never experiences the sensor disappearing from one side of the working space to reappear on the other side. This would be the case if the working space was the same size or larger than the environment.

---

> ### Author Response · Authors · 2018-11-22
> **Reply to reviewer 1 - part 3**
>
> As you noted, the spatial representation can only emerge if the data the agent has access to is “rich enough”. Nonetheless, this richness is not about the complexity of sensorimotor data, but about the fact that it should contain the types of sensorimotor invariants that space induces.
>
> - Details:
> Sec 2: Regarding (Jonschkowski and Brock, 2015), we didn’t mean to target a particular prior. Many of the different “robotics priors” used in this work are directly or indirectly related to space. This includes the “simplicity” prior (the space in which the task is defined is of low dimenion), the “temporal coherence” prior (the dynamics of the system is assumed to slowly move it in space), or the “proportionality” prior (the amplitude of the commands is assumed to be related to the amplitude of the displacement). This work is very interesting, and the first version of the Related work section might have appeared too critical. We simply highlight in this work that such priors might not be required for the emergence of spatial knowledge.
> The Related work section has been reworked to better present our positioning.
>
> Sec 3: As mentioned earlier, motor state means posture (position of each joint) in our formalism. But it could also be seen a proprioceptive reading. Contrarily to a MDP, where m changes the state of the system, here m corresponds directly to the motor state of the agent. It is not an “action”, as is typically considered in RL. Note however that the full “state” of the agent is defined by considering both the motor state m and the sensory state s together. In this formalism, many components of what would traditionally be considered part of the agent’s state is instead included in the state of the environment, ε.
>
> The sentence “Given a rigid metric space: when the agent moves around the same type of movements lead to the same kind of transformations independently of where the agent is located” appears to capture part of the argument we put forward, but it could be interpreted in different ways. In order to clarify it, we can summarize the logic as follows. We assume that the external space in which the agent and its environment are embedded is a rigid Euclidean metric space. When the agent changes its motor state, it changes the position of its sensor in an egocentric frame of reference in the external space. Some of the movements it produces are equivalent in the sense that they correspond to the same displacement (vector) in the external space. However, these equivalences are not directly accessible to the agent. It has only access to raw sensory data and raw motor data, which both differ from the actual position of the sensor in the world. Two identical external displacements of the sensor (for instance moving 5 cm upward when starting from the left of the agent or from just in front of the agent) appear initially as two different motor transformations. Nonetheless, we show that when the environment can move, different motor changes which correspond to the same external displacement can generate the same sensory changes. The agent can thus discover they are equivalent. Moreover, this property holds for any state of the environment (position and content).
> We significantly reworked the Problem setup to better present these ideas and the formalism we use to describe them. In particular, equations are used to define the sensorimotor invariants in the new version of the manuscript.
>
> Sec 4.2: We followed your recommendation and rephrased the description of the three types of exploration. The MTM type is indeed similar to having randomly selected sensorimotor pairs (m_t, s_t) from the interaction, except that we make sure that no two successive pairs correspond to the same state of the environment.
> The difference between the MTM and MMT types is that, in the former, the environmental change happens *between* the two consecutive pairs (m_t, s_t) & (m_{t+1}, s_{t+1}) of a sensorimotor transition, while in the latter, the environmental change happens *after* the two consecutives pairs have been collected.
> In practice, in the MMT case, the agent produces 100 sensorimotor transitions after each displacement of the environment. However, because some motor states might not be viable, the number of sensorimotor transitions collected for each position of the environment might vary between 0 and 100. Moreover, each displacement of the environment corresponds to a instantaneous jump.
> The different types of exploration, as well as the precise way the sensorimotor data is collected have been clarified in both the Experiment section and Appendix of the revised manuscript.

---

> ### Author Response · Authors · 2018-11-22
> **Reply to reviewer 1 - part 2**
>
> We added a comment in the manuscript regarding the future release of the code used for the simulations and network optimization. This code is currently being reworked to improve its modularity, so that other researchers could easily test different kinds of agents and environments. Our code is also partially based on the Flatland simulator (Casselles-Dupré et al., 2018) which hasn’t been released yet. We thus need to wait for its official release at the end of the year before making our code available to everyone.
>
> - Discussion:
> No assumption is made in the approach regarding the complexity of the environment, or of the sensory apparatus - although of course it hasn’t been evaluated on more complex setups yet. The spatial invariants we describe are based on notions of equality and neighborhood, which hold regardless of the complexity of the sensory states themselves (and indirectly of the content of the environment). Non-linear distortions induced by the sensor are not an obstacle to the approach, assuming they do not destroy the continuity of the sensory experience (see for instance “Learning diffeomorphism models of robotic sensorimotor cascades”, A.Censi et al., 2012, or “Unsupervised model-free camera calibration algorithm for robotic applications”, G.Montone, 2015, which follow a similar approach).
> As mentioned in the conclusion, an assumption has however been made regarding the agent-environment interaction: only translations of the sensor and of the environment have been considered so far. The problem of metric regularization is indeed less trivial with rotations, and hasn’t been fully formalized yet. In particular, even if one can regularize the metric associated with rotations, it is difficult to represent them in a Euclidean space (like the representational space in this paper) due to the manifold defined by the rotational metric being looped. This is a problem we plan to investigate in the future.
>
> We agree with you that it is not obvious that a spatial representation can emerge without any pressure for minimality or sparsity. However, this work shows through simple examples that the pressure from accurate sensorimotor prediction seems sufficient. This is the reason why we find these results very exciting and relevant for the Representation Learning community.
> The intuitive explanation is that by capturing the sensorimotor invariants induced by space, the network increases its prediction capacities. Indeed, by nature, these invariants describe how some sensorimotor relations are always true, regardless of the state of the environment. By building a representation which takes into account these relations, the agent gets the ability to generalize to new experiences. The simplest example of this phenomenon is as follows: Imagine the agent has been through multiple environmental states, and discovered that m_i and m_j have always generated identical sensory states. By representing m_i and m_j the same way, it gains the ability to extrapolate to new situations. For instance, the next time the agent is in an unknown environmental state, it can predict the sensory state associated with m_j, simply by knowing what sensory state m_i generates. A similar argument can be made for equivalent motor changes (m_i -> m_i’).
> Nonetheless, we agree with you that, in theory, the network should be able to learn a sensorimotor mapping, even if the motor representation does not capture spatial properties. However, it seems like this does not happen in practice. The structured representation appears to be a strong attractor during convergence. We plan to further investigate this question in the future by looking for the exact conditions of this convergence.
>
> The formalism we use indeed assumes a direct control of the sensor position by the motor state. This is a debatable working hypothesis, as one could easily argue that other systems are dynamically controlled. As clarified in the new version of the manuscript, the motor state m can rather be seen as a proprioceptive information (Philipona et al., 2003). In case no proprioception is available, some mechanism would be required to integrate the motor commands and estimate a posture. This is for instance this kind of approach which is proposed in (Ortiz & Laflaquière, 2018), by using a recurrent neural network which learns to perform this integration.

---

> ### Author Response · Authors · 2018-11-22
> **Reply to reviewer 1 - part 1**
>
> We would like to thank you for your rich feedback. We have revised the manuscript to improve the problem description, the mathematical formalization, and the simulations description. More generally, we significantly modified the paper to improve the write-up, and to clearly introduce concepts that appeared too vaguely defined in the first submitted version.
> We answer your additional questions in detail below.
>
> - Strong aspects:
> We thank you for your positive feedback regarding the relevance of this work for the Representation Learning community.
>
> - Weak aspects:
> We reworked the Problem setup section, and the Experiments section to better describe the three different kinds of exploration, the two types of spatially-induced sensorimotor invariants, and how they relate to one another.
>
> The formalism we use indeed differs from the POMDP formalism usually used in the Reinforcement Learning framework. It considers a sensorimotor mapping s = φε(m), where m corresponds to a motor configuration (posture), or to proprioceptive readings. It is directly inspired by Poincaré’s description of the problem of space perception (Poincaré, 1895), as well as its latter formalization based on differential geometry (Philipona et al., 2003). Although not a standard, this type of formalization has been used in various papers applying the Sensori-Motor Contingencies Theory to different perceptive problems. A similar formalism can also be found in the literature related to forward/inverse model learning, or even to goal babbling. It has the benefit of leaving aside the dynamics of the system (which can be dealt with using additional control modules), although this is of course a strong working hypothesis.
> We reworked the Problem setup section to better introduce this formalism and the problem we address in this work.
>
> The comparison with other baselines is a tricky question that we considered when writing the paper. Beyond the page limit consideration, it is not clear how this work can be compared to other approaches. Indeed, many approaches exist in which spatial structure is enforced by introducing priors in the learning system. By construction, these approaches would then lead to similar qualitative results. A comparison would then be rather pointless, beyond merely stating that these priors play the role they have been designed for. On the other hand, other less supervised approaches tackle intrinsically spatial tasks, but do not explicitly build any spatial embedding to which we could compare ours.
> More generally, the aim of this work is not to propose an optimized way to tackle a spatial problem. We rather focus on core intrinsic mechanisms which could lead to the emergence of spatial knowledge in a naive agent. In such a context, we doubt that comparing the performance of algorithms of different natures (different mechanisms) would better convey our message.
> We however considered one meaningful baseline in the paper by testing the MTM type of exploration. These simulations rely on the same fundamental mechanisms (sensorimotor prediction) as the other simulations. However, they correspond more to what a non-situated agent would produce in a typical supervised or unsupervised learning setup, where the spatio-temporality of the data is not respected. The representations produced in the MTM case are thus used as a baseline to study how other types of exploration influence the motor representation.
>
> In this work, we indeed assume and show that spatial knowledge should emerge from the motor space. This is an argument that has initially been made by Poincaré, and was later more formally presented in (Laflaquière et al., 2015). The intuition beyond this claim is that we do perceive space as a stable structure. Yet, sensory experiences vary greatly, and in particular way more than the underlying spatial properties of our experience. For instance, the sensory experience associated with a single position in space changes when the environment changes. On the other hand, the motor experience exhibits a strong relation with spatial properties. For instance, the position of a sensor in space is fixed for a fixed motor posture. The motor space thus appears to be a stable bedrock on which to build a stable notion of space.
> Note however that in order to discover the structure of space, both motor and sensory experiences are necessary. Despite being grounded in the motor space, the spatial representation built in this work are based on both motor and sensor data. Intuitively, one could say that the sensory flow defines the way the motor information should be transformed to capture spatial invariants.

---

### Official Review · AnonReviewer3 · 2018-11-03

**Rating:** 4
**Confidence:** 4

**Review:**

The paper investigates the exploratory conditions under which spatial representations will emerge as a byproduct of learning to predict the next sensory observation. In particular, the authors test three exploratory conditions:
(i) When the set of sensory-motor interactions (st, mt, s_{t+1}, m_{t+1}) are inconsistent.
(ii) Environment is static and set of sensory-motor interactions are consistent.
(iii)  Environment is dynamic and the set of sensory-motor interactions are consistent.

Authors measure the quality of learned spatial representations by computing disparity between the set of agent positions and the corresponding embedding of motor commands learned by predicting the next sensory observation. They conclude that under condition (iii), the disparity is minimum -- i.e. the agent is best able to discover the spatial structure of it environment.

Philosophically, I love the direction of this work -- understanding the origins of our spatial representations. But, I am concerned with the delivery. My concerns/questions are as following:

(a) Firstly, the writing is too verbose and vague without clarifying the details and there are too many references to Laflaquiere et al. I would recommend the authors to be more precise, i.e. define topological/metric invariants, clarify how in-consistent sensory/motor pairs are sampled in MTM condition and how environment is perturbed in MMT condition. These things are defined at a high-level and not precisely.

(b) The whole premise of comparing the embedding of motor commands and the agent's spatial configuration only works under a special condition -- s = φε(m) (section 3 of the paper), which is not general. For e.g. if an arm is “torque controlled” it is not possible to predict the location of the arm (i.e. a potential sensory observation) just from the torques. Additional knowledge of the agent’s state such as current position and the velocity of the arm is required. In the examples mentioned in the paper, the arm is “position” controlled, i.e. given the orientation of each joint (i.e. the motor command) it is possible to predict the sensory observation.  This is a very special case. In biology for example, we control the flexing of muscle fibers using the motor system, we can’t directly output positions of the arm. The general, condition of operation should be: s_{t+1} = φε(m_t, s_t).

(c) Authors argue that in order to learn metric invariants, the agent needs to observe the same sensory state under different motor commands. They further argue that in the MMT condition, where the environment also translates, this affect is achieved and therefore metric invariants are learned. My position is that this is simply an artifact of the restricted problem setup where  s = φε(m) holds. In a more general setup, s_{t+1} = φε(m_t, s_t) there is no requirement for the environment to move. An arm with different torques can be at the same position and hence the condition imposed by authors should be specified naturally even in MM environments. What do the authors think?

(d) Under the condition of,  s = φε(m) difference between MM and MMT appears that in one case (MM), neural network is trained without translation perturbations, and in other case MMT is a form of data augmentation with translation perturbations. I am not sure if there is any other justification for why only topological invariants should be learned with MM and metric invariants with MMT. To me it seems like training with data augmentation leads to better metric learning. Do the authors have any other insights — I would love to know.

(e) Finally the embeddings are useful, if they are useful for an end-task. I would love, if the authors evaluated the learnt embeddings in each of the three conditions for some end tasks such as reaching in case of an arm or something else that is more feasible.

Despite it being a very interesting topic, due to theoretical concerns outlined above, I cannot recommend the paper for acceptance. With a strong rebuttal it is possible to convince me otherwise.

---

> ### Author Response · Authors · 2018-11-22
> **Reply to reviewer 3 - part 3**
>
> (e)	We agree with you that an obvious use of the motor representation would be a reaching task. This is an application we did not include in this paper, primarily due to space limitations. We also wanted to focus our message on the fact that spatial knowledge can be acquired without the need for priors or for a particular task/reward. However, application of the representation to reaching have already been proposed in (Laflaquière et al., 2018). They show in particular that capturing the metric regularity allows the agent to generate optimal straight trajectories in the external space.
>
> We significantly reworked the manuscript to take into account all reviewers’ feedback. In particular, we clarified the formalism and improved the wording, which was often too verbose or vague. We hope that the paper is now more compelling and better delivers its message, which we believe to be of interest for the Representation Learning community.

---

> > ### Comment · AnonReviewer3 · 2018-12-10
> > **Response to Rebuttal**
> >
> > Thanks for significantly revising the manuscript. It's much easier to follow and the question the paper seeks to address is clearer. I have following comments:
> >
> > (b) A general definition of a motor state can encompass proprioceptive readings, but these readings are not directly controllable. This means the agent cannot modify them at will without learning a mapping between its true motor commands and proprioceptive readings. In theory, this is possible. So more than learning a mapping between the motor commands and the positioning of sensors, it seems like the propose formulation leads to learning of mapping between robot's postures and the placement of sensors. Is this a correct conclusion? If yes, the claims of the paper should be revised to reflect this.
> >
> > (c) I am not against the claim that spatial representations cannot emerge from sensorimotor interactions, in fact I believe in it. I can see your line of argument better now -- but the following is still unclear to me -- doesnt the environment need to move in a very special way so that the two motor transitions (mi-->mi') and (mj-->mj') that have the same external displacement will also see the same sensory transitions? Let (mi-->mi') happen at time t_i and (mj-->mj') happen at time t_j. Environment might have changed significantly between t_i and t_j -- so how is this claim valid?
> >
> > (d) Your line of argument is clearer to me. However, I would appreciate control experiments with translational data augmentation.
> >
> > (f) Another thing, I am unclear about is -- how is the interaction data generated to train the various models? Is it via random interaction with the environment?
> >
> > Given the questions above, I am not inclined to change my ratings at the moment.

---

> > > ### Author Response · Authors · 2018-12-10
> > > **Additional reply to reviewer 3 - part 2**
> > >
> > > d) We could definitely run some control experiments with translational augmentation; although they couldn’t be added to the paper now that the updating deadline is passed.
> > > We’re not sure however about the type of experiment you have in mind. Would you add translational noise on the motor states?, on the sensor position?, on the sensor states?, or on the environment position? It seems to us that adding translational noise on the sensor position or on the environment position would be equivalent to the MTM type of exploration we considered. Adding noise to the motor state wouldn’t change the outcome of the learning either, as the exploration of the motor space is already random. Finally, adding translational noise on the sensor states seems to be more common. However, it usually makes sense when using convolutional layers, which are a priori structured to be invariant to translation, and in a more general purely sensory pipeline (typically, an image classification task). In our case, we would first need to determine how the sensory input should be changed when translated, and in particular the parts of the sensory input for which no information is available in the original input (people for instance use inpainting techniques when dealing with images). This means creating additional sensory inputs that the agent actually never experienced in its “life”. Then, because our architecture is sensorimotor, we expect the effect of the translational noise to be heavily destructive. Indeed, even in the MTM type of exploration, the mapping between a motor state m and the associated sensory state s is deterministic. Given the continuity assumption we make, the network should thus be robust to a small amount of noise on the sensory (and motor) data. But for any significant translational noise, the network wouldn’t be able to even learn the simple m -> s mapping. We don’t expect any structure to be captured in this context.
> > > Is this what you have in mind? Or do you have any other suggestion regarding this translational augmentation, maybe by considering another network architecture overall?
> > >
> > > f) In the current implementation of the simulation, the interaction with the environment is indeed random. It prevents any bias we could introduce by designing a specific policy for the agent. For instance, we do not explicitly ask the agent to look for motor transitions which would have the same sensory outcome. The second half of section 4.2 is dedicated to the description of the data generation, which varies slightly depending on the type of exploration considered (MTM, MM, or MMT). Further details are also provided in Appendix A.

---

> > > > ### Comment · AnonReviewer3 · 2018-12-10
> > > > **response**
> > > >
> > > > Thanks for the clarifications.
> > > >
> > > > (b, c): From the discussion it seems to me that the main message of the paper is: it is possible to learn about the spatial structure of space by predicting future sensory observations. The spatial structure refers to placement of sensors in the internal coordinate of the agent (and not concepts like 3D structure of the world etc.). There are two situations:
> > > >     (i) If only the agent moves, then one can only learn topological regularities.
> > > >     (ii) If the agent + environment moves, one can learn about metric regularities as well.
> > > >
> > > > Do the authors agree that this is the core claim? If so, if I recommend the paper for acceptance, will they modify the text to clearly state this (especially the definition of spatial structure)?
> > > >
> > > > The reason for (ii) is that for change in the environment, there exist multiple motor commands that lead to the same (future) sensory observation (because multiple displacements can be compensated by multiple motor commands) -- however, this would probably confuse the agent also. There are two sources of redundancies -- one due to an overcomplete motor system -- which would cause two motor commands to map to the exact same sensory observation (given static environment). In case of dynamic environment, the same commands can lead to different sensory observation. If we consider, motor commands mi, mj that are not equivalent due to motor redundancy -- they could lead to same or different sensory observations depending on how the environment moves. Furthermore, because the sensors are sampling the world, there are different world states that can correspond to the same sensory observation too (but lets ignore this for now).
> > > >
> > > > So intuitively, when the environment is not moving, it is easier for the agent to reason about which motor commands are equivalent to each other and which are not. In case the environment is moving, these equivalences should be harder to capture right? Maybe I am not undertstanding your line of argument -- but if there is a concrete mathematical description in any previous works -- please refer to me to the exact section, where such an argument is made.
> > > >
> > > > Also, I donot agree that there exist a motor association for each change in the environment. For e.g. the agent might only move in (x, y), whereas the object might move in the Z-direction. Under your framework, would the agent discover the 3D structure of the environment or would it only know about the 2D structure?
> > > >
> > > > (d, f) -- I am satisfied with the responses. The translation noise experiments would make more sense when the inputs are real images.

---

> > > > > ### Author Response · Authors · 2018-12-10
> > > > > **Additional reply to reviewer 3**
> > > > >
> > > > > Thank you for your quick feedback.
> > > > >
> > > > > We agree with your description: “it is possible to learn about the spatial structure of space by predicting future sensory observations. The spatial structure refers to placement of sensors in the internal coordinate of the agent.” However, we would also argue that this emergence of an egocentric notion of spatial configuration of the agent is the bedrock for the development of richer notions of space (including the spatial configuration of objects around the agent). This is not a line of argument we develop in this paper, but multiple previous papers have done so, including the original works by H.Poincaré and J.Nicod ((Poincaré, 1895), (Nicod, 1924), (Laflaquière et al., 2018), (Terekhov et al., 2016), not to mention all of them). Indeed, the sensor configuration directly inherits its structure from the spatial structure in which it is defined. Moreover, the mechanism which allows the agent to discover the spatial metric is based on the existence of “sensorimotor compensation”, a phenomenon that H.Poincaré claimed to be the foundation of the concept of space as such (see (Poincaré, 1895)). We do not develop this philosophical stance in this paper, due to the lack of space and more applied machine learning orientation of the results, but you can refer to (Laflaquière et al., 2018) for a more complete discussion on the topic.
> > > > >
> > > > > Everything you say about considering how motor states can be related to varying (or not) sensory states depending on the environment is correct. And as you point out, some motor-sensory states associations might be difficult (or even impossible) to discover when the environment moves. This is precisely why we do consider a sensorimotor predictive task, where the building block is a *transition between two sensorimotor states*, and not simply isolated sensorimotor states. Beyond the philosophical argument that an acting agent should be interested in the way it can *transform* its sensorimotor experience, considering such transitions allows the agent to use the first sensorimotor state as a reference to determine how the next motor state will map to a sensory state. In that way, changes in the environment do not disrupt the agent’s ability to map a motor state to a sensory state (as long as they do not happen during the transition).
> > > > >
> > > > > You are also right to point out that they are two kinds of redundancies in the system: one due to a potentially overcomplete motor-position mapping, and another one which can be interpreted as a redundancy between the sensor position and the environment position. This is precisely this second kind of redundancy that Poincaré named “sensorimotor compensation”: because there is such a thing as space, both the agent and its environment can undergo the same kinds of displacements. It’s like the overall agent + environment system is itself overcomplete, due to space.
> > > > > The first kind of redundancy (overcomplete motor) can be discovered as long as the agent can experience the sensory equivalence of different motor states (coined “condition I” in the paper). Or said otherwise, as long as the agent can produce motor changes which do not change the sensory input, in a temporally static environment. The second kind of redundancy is similar but involves both the agent and its environment. It can thus be discovered only if the environment moves (coined “condition II” in the paper). This is these redundancies that we show are captured by the predictive neural network, although we do name them "invariant" instead of "redundancy" in the paper.
> > > > >
> > > > > We could of course slightly revise the problem description section to better define what we mean by “spatial structure” if the Chairs allow it.
> > > > >
> > > > > Finally, your last remark is perfectly correct. In a scenario in which the agent can only move its sensor in 2D, while its environment can move in 3D, we claim that the agent would develop a 2D notion of space. Any change of the environment along the 3rd dimension would be interpreted as a change of nature instead of a change of position. This would be similar to what happens when we, humans, look at the rotation of a 4D hypercube; it appears to change its shape, instead of its orientation. This link between the ability to move and the perception of space goes back to Poincaré’s initial papers, where he considers, on the contrary, agents that would perceive a 4D space. This idea has since been studied in numerous papers about the perception of the dimension of space (see (Philipona et al., 2003), (Laflaquière et al., 2012), or (Valentin Marcel et al, 2018)).

---

> > > ### Author Response · Authors · 2018-12-10
> > > **Additional reply to reviewer 3 - part 1**
> > >
> > > We thank you for your additional feedback on the paper. We answer your different questions in details below.
> > >
> > > b) As you noted, the mapping learned by the agent can be seen as a (robot posture) -> (sensor position) mapping. It is however a bit more general in principle. Indeed, we don’t assume that the state m directly corresponds to the robot posture. We only assume that there exists some static relation between m and the robot posture. As a consequence, m can have a different structure (topology and metric) that the actual robot posture. For instance, m can be of greater dimension than the actual posture, similarly to how numerous muscles can co-determine the position of a single joint. The static aspect of this relation between m and the posture is thus what is relevant (see also (Philipona et al., 2003) for a related discussion). As described in the paper, the state m can thus be seen as a motor command statically controlling the robot’s posture, or as a proprioceptive reading related statically to the robot’s posture.
> > > The deadline for uploading a new version of the manuscript for review is now passed; but we could emphasize the importance of this static hypothesis even more in the Problem Setup section in a final version of the paper.
> > > In case the robot is dynamically controlled, it is true that there isn’t a direct relation between the motor command and the proprioceptive (posture) readings. But as you noted, it is also true that the relation between the two can be learned. As mentioned in the Related Work section, and in our previous reply, this has in particular been studied in (Ortiz et al, 2018). In this paper, we thus assume that this problem is solved, or orthogonal to the core message of this work: space induces structure in the experiences gathered by a situated environment, and this structure can be naturally captured when trying to predict future experiences. Including an additional mechanism to integrate motor commands in this paper would not be possible given the limit on the number of pages, and would also obfuscate the message by adding modules in the neural architecture which are not directly related to it.
> > >
> > > c) We’re glad that the description of the approach in the revised manuscript is now clearer.
> > > We took the example of two motor transitions (m_i->m_i’) and (m_j->m_j’) in our previous reply because it was convenient to outline the mechanism underlying the discovery of the spatial metric regularity. However, once this simpler mechanism is understood, starting the reasoning from the movements of the environment gives a more interesting perspective on the problem.
> > > To answer your question, when specifically considering (m_i->m_i’) and (m_j->m_j’), the environment indeed would need to move in a very special way: between the two transitions, it would need to undergo a displacement equal to the displacement that exists between m_i and m_j (see (Laflaquière et al, 2018) for an extensive description and illustration of this idea). Of course, precisely undergoing this displacement is unlikely. Fortunately, the continuity hypothesis (which seems confirmed experimentally) we formulate about the sensorimotor mapping is such that the structure can still be captured if the displacement of the environment is “close enough” to the precise displacement it should be in this context.
> > > Nevertheless, it is actually more interesting to consider a dual perspective. Instead of considering two motor transitions, let’s consider a single environment displacement. For any transition that the agent experienced before the environment moved, there exists another transition which is equivalent to it from a sensory perspective after the environment moved. If we consider that the motor space can be sampled with infinite precision, there exists a continuum of such motor associations for each displacement of the environment. In practice, there is obviously only a limited set of motor states experiences by the agent. Yet, thanks to the continuity hypothesis, each displacement of the environment can actually induce a large number of equivalences that the agent captures.

---

> ### Author Response · Authors · 2018-11-22
> **Reply to reviewer 3 - part 2**
>
> (c)	Our working hypothesis is indeed that a naive agent can discover the metric regularity of the external space by exploring an environment which can move. It can then experience the same sensory changes for different motor commands, which indicates that these motor changes are equivalent in the external space.
> We do not think that this line of argument is an artefact of the way we formalize the sensorimotor mapping, but rather a more profound consideration regarding how the concept of space can emerge from raw sensorimotor experiences (Poincaré, 1895). We seriously took your suggestion regarding a torque control setup into consideration. However, we do not see how this could help discovering metric regularity invariants. As you correctly noted, there wouldn’t be a direct mapping between motor commands and spatial configurations (positions) in such a system. Multiple torques can be associated with the same position, and multiple positions can be associated with the same torques. Yet, our objective is to build an internal representation of the sensor position in an egocentric frame. Some mechanism would thus be required to integrate these commands and derive a position (see reply to comment (b)). Now, let’s assume that such a mechanism exists, or that the agent has access to some proprioceptive information (posture). When exploring a static environment, a specific sensory change is associated with each posture change. The agent thus has no way of “comparing” the different posture changes beyond the topological invariants we describe in the paper. Let’s assume that two motor changes (m_i -> m_i’) and (m_j -> m_j’) actually correspond to identical displacements (same vector) in the external space. They are a priori associated with two sensory changes (s_i -> s_i’) and (s_j -> s_j’) which are different. The agent thus has no incentive to assume they are equivalent, as all motor changes are associated with different sensory changes anyway. On the contrary, if the environment moves around, the agent can experience the same sensory transitions (s_i -> s_i’) and (s_j -> s_j’) with the two motor changes (m_i -> m_i’) and (m_j -> m_j’). However, other motor changes, which correspond to different external displacements, cannot. The agent thus as a way to compare the motor changes and assess which ones are equivalent.
> We modified the Problem setup section to better describe the topological and metric invariants considered in this work. We now use clearer mathematical notations, and provide a better description of the phenomena which induce them.
>
> (d)	As mentioned in our reply to comment (c), only topological properties can be captured if the environment stays static. In such a case, each motor state m_i is associated with a specific sensory state s_i, and a fortiori each motor change (m_i -> m_i’) is associated with a specific sensory change (s_i -> s_i’). The only “metric” comparison the agent can perform between its motor changes is thus to look at the amplitude of the different sensory changes (s_i -> s_i’). However, this amplitude is not a direct characterization of the amplitude of the associated external displacement. Only topological properties hold (working hypothesis): infinitesimal displacements of the sensor are associated with infinitesimal sensory changes. As a result, the representation built by exploring the sensory manifold can have an arbitrary metric where nothing enforces identical external displacements to be represented with identical vectors. This result is confirmed by our simulations.
> As described in our reply to comment (c), adding displacements of the environment allows the agent to experience the equivalence of different motor changes. This is also confirmed by our simulations. This addition is not simply akin to adding noise or translation-based data-augmentation. Translation-based augmentation is generally used in conjunction with convolutional networks, and only limited to the “field of view” of the sensor, which is not the case here. If one really wanted to frame it as data augmentation, it would correspond to a very specific type of augmentation which respects the spatial invariants that we want the agent to be exposed to.

---

> ### Author Response · Authors · 2018-11-22
> **Reply to reviewer 3 - part 1**
>
> We would like to thank you for your constructive feedback. Based on your comments, we revised the manuscript to improve the description of our theoretical framework, as well as the presentation of the simulations and results.
> We answer your comments and questions more in detail below.
>
> (a)	We agree with you that the first version of the paper was relatively verbose, and sometimes vague as a consequence. The theoretical considerations and associated math on which this work is based can easily take a lot of space. In order to respect the page limit, our original motivation was to propose high-level textual descriptions of these considerations. As you and other reviewers rightfully noted, this turned out to be too vague and confusing.
> In the revised manuscript, we therefore decided to reduce the use of heavy descriptive sentences, and to rely on concrete mathematical formalization instead. This is particularly visible in the Problem setup section, which we significantly extended. We slightly shortened the Results section as a consequence, without modifying its core message. We also removed references to (Laflaquière et al., 2018) when possible, and better explained the sampling method for the three types of exploration, both in the main test and in the appendix.
>
> (b)	Our formalism indeed assumes a “static” mapping s = φε(m), which is different from how the problem of spatial knowledge acquisition is usually formalized in the reinforcement learning framework. This “static sensorimotor mapping” approach is inspired by Poincaré’s description of the problem (Poincaré, 1895), as well as its latter formalization based on differential geometry (Philipona et al., 2003). It has since been used in various papers applying the Sensori-Motor Contingencies Theory to different perceptive problems. A similar approach can also be found in the literature related to forward/inverse model learning, or even to goal babbling.
> As mentioned in your comment, it assumes that the system is controlled in position (not in the external space, but in the joint space). This is a debatable hypothesis, as one could easily argue that biological systems are more generally torque controlled. In such a case, two options are available: 1) The motor state m can be seen as a proprioceptive reading (configuration of the joints), instead of a motor command. This approach is for instance discussed in (Philipona et al., 2003). 2) The motor state controls the dynamics of the robot, and some integration mechanism needs to be introduced in order to estimate the posture of the robot. This is for instance this kind of approach which is proposed in (Ortiz & al., 2018), by using a recurrent neural network which learns to perform the integration.
> The Problem setup section has been modified to better explain the nature of the motor state m.

---

### Author Response · Authors · 2018-11-22
**Reply to all reviewers**

We thank all three reviewers for their careful reading and constructive feedback. They have helped us improve many sections of the manuscript. We took into account all reviewers suggestions, and answered their questions in each individual reply. Major revisions to the paper include: an extension of the Related work, a careful rewriting of the Problem setup which better formalize spatial invariants, a clarification of the sampling method for each type of exploration, and an overall clarification of the wording used throughout the manuscript. We have uploaded the new version of the manuscript, so that the three reviewers can provide additional feedback, if needed.

---

### Meta-Review · Area_Chair1 · 2018-12-14
**uncertain link between inferring sensor locations and spatial structure**

**Confidence:** 2
**Recommendation:** Reject

**Metareview:**

This paper is borderline for publication for the following reasons:
1) the title is misleading. The majority of the ICLR audience understands by "spatial structure" the structure of the external 3D world, as opposed to the position of the sensors in the internal coordinate system of the agent. Though the authors argue that knowing the positions of the sensors eventually leads to learning the 3D world structure, this appears like a leap in the argument.
2) The equation s=\phi(m) described a mapping from robot postures to sensory states. This means the agent should remain within the same scene. The description of this equation in the manuscript as "The mapping  can be seen as describing how “the world” transforms changes in motor states into  changes in sensory states ..." makes this equation appear more general than what it is. s'=\psi(s,m) would be better described by such sentence.

---

> ### Author Response · Authors · 2018-12-21
> **Response to metareview - part 2**
>
> 2) The question of the mapping s=\phi_{\epsilon}(m) and, more generally, of the formalism we use has been discussed in details with AnonReviewer1 and AnonReviewer3 (see "Reply to reviewer 1 - part 1", "Reply to reviewer 3 - part 1", and "Additional reply to reviewer 3 - part 1"). We will not go through the whole justification here again. However we would like to point out that this formalism, although different from the one commonly used in the Reinforcement Learning framework, is not a contribution of this work. It has already been introduced in previous works (see in particular (Philipona et al., 2003)), and such a sensorimotor-mapping approach is often used in some fields of Robotics and Machine Learning related for instance to forward/inverse models, motor/goal babbling, or sensorimotor approaches of perception such as the SMC theory. As discussed with the reviewers, we agree that the relevance of this formalism with respect for instance to neurobiology can be discussed. As any formalism, it relies on assumptions that can be debated, and this is why we were careful to use the subjunctive mood and the verb "assume" when introducing it. However, contrary to what you suggest, it is not in principle less general than a different mapping of the kind s'=\psi(s,m). Indeed, in your description, you omitted the environmental state \epsilon which parameterizes the sensorimotor mapping \phi. Its role is equivalent to the one of "s" in the formulation s'=\psi(s,m). Conceptually, the relation between the environment and the world which leads to the sensation s is considered to be part of \epsilon. Imagine for instance that an agent with a fixed posture m can be placed in multiple locations in the environment, generating different sensations. From an egocentric and sensorimotor perspective, this can be seen as the agent always being in the same posture, but the environment changing its state \epsilon (in this case, its location relative to the agent). Similarly, the environment (or "scene") can also change. The rationale behind this formal choice is that s depends on the state of the environment, which the agent does not control directly. As a consequence, one could propose to rewrite s=\phi_{\epsilon}(m) as s'=\phi_{\epsilon}(s, m), but s and \epsilon would be redundant.
> Another difference between this formalism and the one commonly used in RL setups, is that we consider m to be a static motor state (posture, or proprioception) instead of a dynamic action. This point has also been discussed in particular with AnonReviewer3. We only make the assumption that m and s can be acquired instantaneously (without transient phase) - an assumption that has its own equivalent in the more dynamic RL framework. Moreover, one can pass from one formalism to the other by having mechanisms to integrate and derive the evolution of the motor and sensory states. As mentioned in our previous replies, previous work has already shown that such mechanisms could be learned (see (Garcia-Ortiz et al, 2018)).
> We understand that this static formulation in which the sensorimotor mapping is parameterized by the state of the environment is not usual. Yet, given these different elements and the fact that this formalism is not a contribution of this work, we do not understand why it appears to be so controversial in this work.
>
> We hope that you can take into consideration our reply, as we truly believe that the mechanism we describe in this paper can be of interest for the Representation Learning community.

---

> ### Author Response · Authors · 2018-12-21
> **Response to metareview - part 1**
>
> Dear Area Chair,
>
> Thank you for your feedback, which we considered with great attention.
>
> After careful reading, we are nonetheless puzzled by the two arguments provided to recommend a rejection. These points have indeed already been addressed during the previous discussions with the three reviewers, who - as far as we understand - have been convinced by our additional explanations and modifications of the manuscript.
>
> 1) None of the three reviewers has mentioned any problem with the title during the review process. As mentioned several times in the paper, the work presented here is a practical evaluation of more theoretical developments about the perception of space (in particular (Poincaré, 1895), (Nicod, 1924), (Laflaquière et al., 2018), and (Terekhov et al., 2016)). The references to "spatial structure" in the paper and its title are thus intentional. As already discussed with AnonReviewer3 (see "Additional reply to reviewer 3"), there is a direct relation between the sensor's position captured by the agent and the structure of space in which the agent and the environment are immersed. The topological and metric properties of the representation are directly inherited from the topological and metric properties of space itself. Moreover, the mechanism through which these properties are captured has also already been studied and proposed as the fundamental building block for the perception of space (see the concept of "compensability" in (Poincaré, 1895) and (Philipona et al., 2003)). In addition to these past theoretical developments, we show in this paper how a drive for sensorimotor prediction naturally leads to the capture of the topological and metric properties of the sensor's spatial configuration, which themselves correspond to the topological and metric properties of space ("spatial structure"). Of course, we agree that the representation of the sensor position built by our agents is not a complete characterization of what we, humans, consider as space. For instance, as pointed out in the Conclusion, so far the approach doesn't extend past the agent itself to capture the spatial configurations of objects around it. Yet, based on the previous theoretical works mentioned, we strongly argue that the sensorimotor invariants captured by the agent, and which reflect the topological and metric properties of space, are the foundation for any richer spatial knowledge that a naive agent can acquire. This is why we use the expression "emergence of spatial structure" in the title. (Regarding the question of space as 3D, this approach naturally extends to more dimensions, as shown in (Philipona et al, 2003), or (Laflaquière et al, 2012).)